# Unbiased homeologous recombination during pneumococcal transformation allows for multiple chromosomal integration events

**Jun Kurushima[1], Nathalie Campo[2], Renske van Raaphorst[1], Guillaume Cerckel[1], Patrice Polard[2], Jan-Willem Veening[1]\***

[1]Department of Fundamental Microbiology, Faculty of Biology and Medicine, University of Lausanne, Lausanne, Switzerland; [2]Laboratoire de Microbiologie et Génétique Moléculaires (LMGM), Centre de Biologie Intégrative (CBI), Toulouse, France

**Abstract** The spread of antimicrobial resistance and vaccine escape in the human pathogen *Streptococcus pneumoniae* can be largely attributed to competence-induced transformation. Here, we studied this process at the single-cell level. We show that within isogenic populations, all cells become naturally competent and bind exogenous DNA. We find that transformation is highly efficient and that the chromosomal location of the integration site or whether the transformed gene is encoded on the leading or lagging strand has limited influence on recombination efficiency. Indeed, we have observed multiple recombination events in single recipients in real-time. However, because of saturation and because a single-stranded donor DNA replaces the original allele, transformation efficiency has an upper threshold of approximately 50% of the population. The fixed mechanism of transformation results in a fail-safe strategy for the population as half of the population generally keeps an intact copy of the original genome.

**\*For correspondence:**
Jan-Willem.Veening@unil.ch

## Introduction

The opportunistic human pathogen *Streptococcus pneumoniae* (the pneumococcus) kills over a million individuals each year, despite the introduction of several vaccines targeting its capsule (*Croucher et al., 2018*; *O'Brien et al., 2009*; *Prina et al., 2015*). Because of its ability to take-up DNA from its environment by competence activation, genes associated with capsule biosynthesis are rapidly transferred from one strain to the other thereby contributing to vaccine escape (*Salvadori et al., 2019*). In addition, antibiotic resistance remains a cause of concern and competence-dependent recombination plays an important role in the spread of drug resistance (*Sw et al., 2019*). For example, one of the main genetic sources for penicillin resistance in *S. pneumoniae* is DNA acquired from non-pathogenic Streptococci from the viridans group such as *S. mitis* that also lives in the human nasal and oral cavities (*Bryskier, 2002*; *Janoir et al., 1999*). Consistently, antibiotic-resistant pneumococci and vaccine-escape variants remain an important cause of invasive infections in spite of the introduction of the conjugate vaccines (*Fenoll et al., 2018*; *Levy et al., 2019*; *Ouldali et al., 2018*).

Although pneumococcal competence is one of the best studied bacterial regulatory systems (*Gómez-Mejia et al., 2018*; *Johnston et al., 2014*; *Lin and Kussell, 2017*; *Salvadori et al., 2019*; *Shanker and Federle, 2017*; *Straume et al., 2015*; *Veening and Blokesch, 2017*), and pneumococcal transformation was already discovered in the early twentieth century (*Avery et al., 1944*; *Griffith, 1928*), we have a poor understanding on how competence-dependent transformation drives

pneumococcal population dynamics, serotype displacement and the spread of antibiotic resistance. Importantly, horizontal gene transfer (HGT) via natural transformation is not only conserved in Streptococci but is present in many human pathogens where it promotes the spread of virulence determinants and antibiotic resistance (*Brockhurst et al., 2019*; *Dubnau and Blokesch, 2019*; *Johnston et al., 2014*). For this reason, it is crucial to understand what the main bottlenecks are during the take-up and recombination of exogenous DNA that leads to transformed new genotypes.

In contrast to many other competent pathogens such as *Acinetobacter* spp. and *Neisseria meningitidis* in which competence is constitutively expressed, competence development in *S. pneumoniae* is only activated under specific conditions (*Blokesch, 2016*; *Claverys et al., 2006*). Pneumococcal competence is under control of a two-component quorum sensing system (*Figure 1*). ComC is cleaved and exported by the peptidase-containing ATP-binding cassette transporter ComAB (*Chandler and Morrison, 1988*; *Håvarstein et al., 1995*; *Hui et al., 1995*). Cleaved ComC autoinducer is commonly referred to as competence stimulating peptide (CSP) (*Alloing et al., 1996*; *Håvarstein et al., 1996*; *Håvarstein et al., 1995*). CSP is recognized by the membrane-bound histidine kinase ComD (*Håvarstein et al., 1996*). Once a certain threshold level of CSP has been reached, as the culture reaches higher densities, or when other environmental factors increase local CSP concentrations (*Domenech et al., 2018*; *Moreno-Gámez et al., 2017*), ComD will autophosphorylate and transfer the phosphoryl group to the response regulator ComE (*Martin et al., 2013*). Phosphorylated ComE then dimerizes (*Boudes et al., 2014*; *Sanchez et al., 2015*) and binds to a specific DNA sequence (*Martin et al., 2013*; *Pestova et al., 1996*; *Slager et al., 2019*; *Ween et al., 1999*). The *comCDE* and *comAB* operons are under direct control of ComE, setting up a positive feedback loop. The genes under control of ComE are called the early *com* genes (*Figure 1*). Importantly, phosphorylated ComE also activates expression of the gene encoding the alternative sigma factor ComX. ComX activates transcription of the so-called late *com* genes, which includes the genes required for DNA uptake and integration (*Campbell et al., 1998*; *Dagkessamanskaia et al., 2004*; *Luo et al., 2003*; *Pestova and Morrison, 1998*; *Slager et al., 2019*; *Figure 1*). While regulation of competence is highly diverse between naturally transformable bacteria, the actual DNA uptake and integration machinery is largely conserved (*Chen and Dubnau, 2004*; *Johnston et al., 2014*).

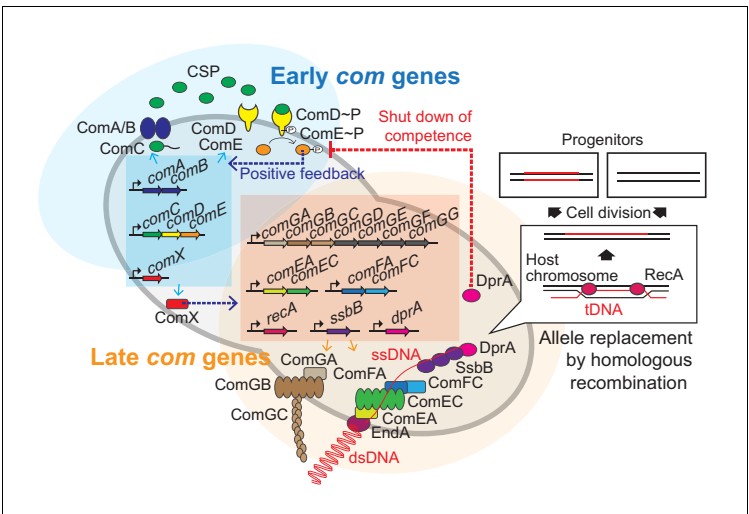

**Figure 1.** Regulation of pneumococcal competence and transformation. Schematic overview showing representative competence-related genes involved in pneumococcal transformation. Competence development is initiated by activation of the early *com* genes (shown in blue area). ComAB exports ComC and processes it into the competence stimulating peptide (CSP). The two-component system ComDE recognizes CSP and positively regulates the early *com* genes. Subsequently, the alternative sigma factor ComX, activates late *com* gene expression (shown in orange area). ComGA, GB, and GC are assembled to form the DNA-binding pilus. EndA is the endonuclease that cleaves dsDNA into ssDNA. ComEA, ComEC, ComFA, and ComFC form the ssDNA uptake channel. Internalized foreign ssDNA is protected by SsbB and DprA. DprA ensures the loading of RecA on single strand tDNA to form a presynaptic filament and the resulting DNA scanning complex is capable of homologous (or homeologous) recombination with the recipient chromosome.

During pneumococcal competence, exogenous double stranded DNA (dsDNA) is bound by a type IV-like pilus (*Laurenceau et al., 2013*) and subsequently sequestered to the DNA uptake machinery (*Figure 1*). Note that in contrast to some other competent bacteria, pneumococcus binds and takes up DNA of any sequence, including non-kin DNA (*Mell and Redfield, 2014*). Next, the dsDNA is processed into single-stranded DNA (ssDNA) by the EndA nuclease and internalized through a membrane pore consisting of ComEC. Once inside, the ssDNA is bound by a competence-specific ssDNA-binding protein, SsbB, and stabilized by DprA and RecA (*Attaiech et al., 2011*; *Bergé et al., 2003*; *Figure 1*). This complex undergoes homology scanning and forms a temporal hetero-duplex during strand invasion which can lead to homologous recombination (*Mortier-Barrière et al., 2007*). The exact details on the kinetics of this process, as well as how the hetero-duplex is resolved in most cells remains elusive. The competent transformation state in *S. pneumoniae* is transient as DprA interacts with phosphorylated ComE to inhibit its activity (*Mirouze et al., 2013*). In addition, several key Com proteins are rapidly turned over after their synthesis, leading to a window of DNA uptake of approximately 15 min (*Liu et al., 2019*; *Tomasz, 1966*; *Weng et al., 2013*).

As most work on pneumococcal competence and transformation has been performed using bulk assays, it is unclear what the actual bottlenecks are during competence development and why one cell will be transformed whereas another one will not. Here, we have set up single-cell transformation assays that allow us to quantify successful recombination events in real-time. This study provides direct evidence for several decade-old models underpinning bacterial transformation, and offers new insights that help explain why competence-induced transformation is so effective in changing global pneumococcal population structures.

## Results

### All pneumococci become competent and bind exogenous DNA

To quantify pneumococcal transformation efficiency and determine at which step potential bottlenecks arise, we systematically analyzed every stage during the process: (1) competence development, (2) production of the DNA uptake machinery, (3) binding of exogenous DNA, and (4) recombination and expression of the newly acquired genetic information (*Figure 1*). While competence development in *B. subtilis* is limited to approximately 10% of the population (*Maamar and Dubnau, 2005*; *Smits et al., 2005*), up to 100% of cells within pneumococcal populations have been reported to become competent when induced with exogenously added synthetic CSP or when grown on semi-solid surfaces (*Bergé et al., 2017*; *Domenech et al., 2018*; *Litt et al., 1958*; *Martin et al., 2010*; *Moreno-Gámez et al., 2017*; *Slager et al., 2014*).

To quantify competence development in clonal pneumococcal populations in a systematic fashion, we constructed a set of reporters. First, we assessed the timing of both naturally induced and artificially induced competence (by the addition of synthetic CSP) at the population level utilizing a firefly luciferase reporter under the control of the late competence *ssbB* promoter (strain DLA3). Cells were grown in C+Y medium at 37°C (see Materials and methods) and growth and luciferase activity were measured every 10 min. As expected, under these experimental conditions, the population rapidly activates *ssbB* in the presence of added CSP, while in the absence of externally added CSP, the *ssbB* promoter peaks after approximately 100 min (*Figure 2A*). To determine which fraction of the cells switch on the competence pathway, we fused the *ssbB* promoter to a fast folding yellow fluorescent protein (msfYFP) and integrated this construct at the native *ssbB* locus (strain VL2219). As shown in *Figure 2B–C*, ~97% of the population was positive for *ssbB* expression 20 min after addition of synthetic CSP as determined by fluorescence microscopy followed by automated image analysis (see Materials and methods for details). Importantly, spontaneous competence without the addition of synthetic CSP was reached in 92% of the population showing that almost all pneumococci, regardless of their cell length and cell-cycle status become naturally competent (*Figure 2—figure supplement 1*).

To test whether competent cells actually produce the machinery required for DNA uptake, we constructed translational msfYFP fusions to three essential components of the transformation machinery: ComGA (ATPase driving the DNA uptake pilus), ComEA (DNA receptor), and ComFA (ATPase driving DNA import) as the only copy integrated at their native locus. After 20 min of

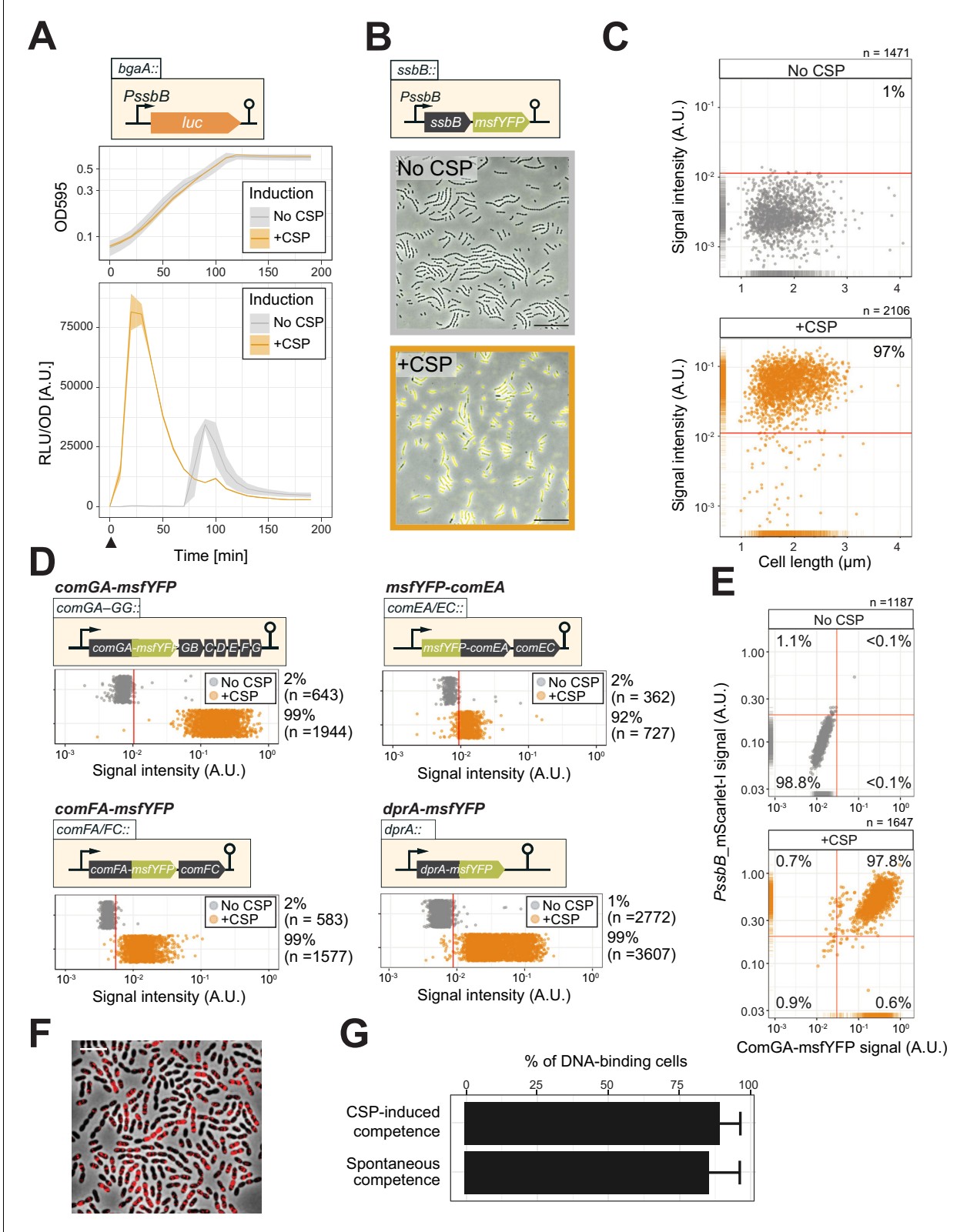

**Figure 2.** Single-cell analysis of competence activation and DNA binding. (**A**) Kinetics of bacterial growth and competence development. Growth curves (top) and OD-normalized bioluminescence activity (bottom) of strain DLA3 (P*ssbB-luc*) in the presence (orange) or absence (gray) of CSP. Arrow indicates the moment after addition of CSP (0 min). Lines and confidence bands represent means of three replicates and SD, respectively. (**B**) Single-cell imaging of fluorescence competence reporter cells. VL2219 (P*ssbB-msfYFP*) was treated with (top, gray frame) or without (bottom, orange frame)
*Figure 2 continued on next page*

*Figure 2 continued*

CSP for 20 min and analyzed by fluorescence microscopy. Images are overlays of phase contrast and YFP signal. Scale bar: 20 μm. (C) Quantification of the imaging. Scatter plots of single-cell YFP signal intensity (y axis) against cell length (x axis), based on microscopy images. Red line indicates the threshold used to score YFP positive cells. Proportion of positive cells (%) is shown. (D) Quantification of cells expressing the transformation machinery. Fluorescence signal intensity for indicated strain harboring *comGA-msfYFP* (VL2536), *msfYFP-comEA* (VL2537), *comFA-msfYFP* (VL2538) or *dprA-msfYFP* (VL3355) treated with (orange) or without (gray) CSP for 20 min. Red line indicates threshold for YFP positive cells. Proportion of positive cells (%) is shown. (E) Correlation between competence activation and ComGA production. VL2536 (*comGA-msfYFP*, P*ssbB_mScarlet-I*) was incubated with or without CSP. Scatter plot of single-cell YFP signal intensity (translational fusion of ComGA) against mScarlet-I signal (transcriptional fusion to *ssbB*). Red line indicates threshold used. Proportion of positive cells (%) is shown on each plot. (F and G) DNA-binding analysis using Cy3-labeled DNA added to induced- or spontaneous competent cells. (f) Representative image of Cy3-labeled DNA-bound TD290 (*ssbB_luc, ΔendA*) cells. Scale bar: 2 μm. (G) Quantification of microscopy images of Cy3-labeled DNA-bound D39V cells. Bacteria were treated with CSP for 20 min to induce competence in C+Y (pH 7.9). Efficiency of spontaneous competent cells (no CSP) is also represented. Total of 6027 cells (without added CSP) and 3082 cells (with CSP) were collected over three independent experiments.

The online version of this article includes the following figure supplement(s) for figure 2:

**Figure supplement 1.** Single cell quantification of spontaneously activated competence.

**Figure supplement 2.** Fluorescence microscopy images of VL2536 (*msfYFP-comEA*), VL2537 (*comFA-msfYFP*), VL2538 (*comGA-msfYFP*), VL3355 (*dprA:: dprA-msfYFP*), and VL361 (*recA::recA-mCherry*) treated with (orange frame) or without (gray frame) CSP for 20 min.

incubation with synthetic CSP, cells were collected for fluorescence microscopy. In line with the fraction of cells that become competent, msfYFP-ComEA, ComFA-msfYFP, and ComGA-msfYFP were also expressed in the majority of the cells (~92,~99, and~99%, respectively) (*Figure 2D* and *Figure 2—figure supplement 2*). A double-labeled strain (strain OVL2536: P*ssbB*-mScarlet-I, ComGA-msfYFP) demonstrated that all competent cells indeed produce the DNA uptake machinery (*Figure 2E*).

Finally, to assess whether the proteins required for recombination and chromosomal integration of exogenous DNA also were expressed in the majority of the population, we constructed translational fusions to RecA and the recombination mediator protein DprA. Similar to the DNA-uptake proteins, RecA and DprA were induced in most competent cells (*Figure 2D* and *Figure 2—figure supplement 2*).

During pneumococcal competence, the capture of extracellular DNA by the ComGC pilus is an essential step for transformation (*Bergé et al., 2002*). To examine which proportion of cells is capable of binding DNA during competence, we labeled extracellular DNA (285 bp *S. pneumoniae* DNA fragment, see Materials and methods) fluorescently with the Cy3 dye. After induction of competence with synthetic CSP of cells mutated for EndA (to prevent degradation of the exogenous DNA),~90% of the population bound extracellular DNA as visualized by fluorescence microscopy (*Figure 2F*). Even without additional CSP, spontaneous competence also led to most cells (89.6%) binding exogenous DNA (*Figure 2G*). As observed before in an unencapsulated R6 strain (*Bergé et al., 2013*), we note that also in the encapsulated serotype 2 D39V strain, DNA mainly bound to the mid-cell positions of the cell, corresponding to the localization of the DNA uptake machinery particularly the ComEA receptor (*Figure 2F* and *Figure 2—figure supplement 1*; *Bergé et al., 2013*). Collectively, these data validate by direct single-cell observations that pneumococcal competence development, the subsequent production of the DNA uptake and integration machinery, as well as DNA binding is highly efficient and occurs in nearly every cell of the population regardless of their cell-cycle state.

## Real-time single-cell analysis of homeologous recombination during transformation

Having established that there are no significant bottlenecks during the process of both induced and natural competence development and DNA uptake, we next set out to develop a system that allows for the direct visualization of successful recombination. Traditionally, transformation efficiencies are evaluated using antibiotic selection methods. However, these selection methods have limitations because they depend on the counting of colony forming units, which can lead to the overestimation of transformation efficiencies, due to inefficient separation of transformed from non-transformed daughter cells and nongenetic inheritance of antibiotic resistance (*Dalia and Dalia, 2019*; *Domenech et al., 2018*; *Ephrussi-Taylor, 1962*; *Ephrussi-Taylor, 1958*; *Figure 3—figure supplement 1*). In order to overcome these concerns and analyze successful recombination events during

transformation at the single-cell level, we developed a fluorescence-based reporter system inspired by a system previously used to observe natural transformation in *S. pneumoniae* (*Bergé et al., 2013*) and other bacterial species (*Boonstra et al., 2018*; *Corbinais et al., 2016*; *Godeux et al., 2018*). To do so, we utilized a fluorescent donor strain in which the gene encoding the abundant histone-like protein HlpA (aka HU) was fused in frame with the gene encoding the red fluorescent protein mScarlet-I integrated at the native *hlpA* locus at 169° on the circular chromosome (*Keller et al., 2019*) (strain VL1780) (*Figure 3A*). A recipient, non-fluorescent strain was constructed (strain VL1784) in which *hlpA* was separated from *mScarlet-I* by a stop codon mutation (G > T) (*Figure 3A–C* and *Figure 3—figure supplement 2A*, *hlpA-stop-mScarlet-I*). Upon uptake, integration and expression of exogenous transforming DNA (tDNA) containing the donor construct (intact *hlpA-mScarlet-I*), successfully transformed recipient cells will produce functional HlpA-mScarlet-I that can be quantified by fluorescence microscopy or flow cytometry (*Figure 3B and C* and *Figure 3—figure supplement 2B*). As this is a recombination event between highly similar but not identical DNA (except for the SNP causing a stop codon), this is called a homeologous recombination event (*Humbert et al., 1995*; *Petit et al., 1991*). Note that this reporter system does not affect growth regardless of the presence of the stop codon (*Figure 3—figure supplement 3*) and that flow-cytometry analysis slightly overestimates the real transformation efficiencies due to cell chaining (*Figure 4* and *Figure 3—figure supplement 4*, see below).

As mScarlet-I is a fast folding red fluorescent protein (*Bindels et al., 2017*), this reporter system should allow for the real-time detection of homeologous recombination during transformation. To test this, we provided competent recipient cells that besides the *hlpA-stop-mScarlet-I* allele also constitutively expressed sfGFP (strain VL1832) with intact *hlpA-mScarlet-I* as donor tDNA in the presence of CSP and then performed time-lapse microscopy (see Materials and methods for details). As shown in *Figure 3D* and *Videos 1* and *2*, recipient cells do not display any red fluorescence in the beginning and then gradually start to express red fluorescence. When quantifying the fluorescence signals and superimposing this on a cell lineage tree constructed using a set of new scripts written in BactMAP (*van Raaphorst et al., 2020*) (see Materials and methods), it becomes apparent that the initial recipient cell already expresses HlpA-mScarlet-I right after the addition of tDNA before the first cell division as red fluorescent signals above background levels can be detected (*Figure 3E*). Notably, only half of the recipients' descendants appear to strongly express HlpA-mScarlet-I (*Figure 3E*, right lineage). Contrary, after three more divisions the non-transformed lineage no longer expresses red fluorescence (*Figure 3E*, left lineage). These results are in line with a recent study in *Vibrio cholerae* that showed a period of nongenetic inheritance in daughter cells during transformation (*Dalia and Dalia, 2019*). Similar observations were made when using a different transformation reporter system (*Figure 5C* and *Video 3*, see below). In line with current models of transformation (*Davidoff-Abelson and Dubnau, 1971*; *Ephrussi-Taylor and Gray, 1966*; *Fox and Allen, 1964*; *Gabor and Hotchkiss, 1966*; *Lacks, 1962*; *Méjean and Claverys, 1984*; *Piechowska and Fox, 1971*), these observations are consistent with a model in which recombination occurs by direct integration of the ssDNA donor and forms a hetero-duplex. Therefore, at least one round of DNA replication and division is required to generate two different homo-duplex chromosomes in progeny cells (*Figure 3F*). The fact that we initially also observe fluorescence in the untransformed lineage suggests that phenotypic expression derived from the acquired allele might occur prior to forming a homo-duplex. In this case, the transformed ssDNA likely replaced the anti-sense, noncoding strand so functional *hlpA-mScarlet-I* could be immediately transcribed after integration via RecA-directed homologous recombination (mismatched pairing between exchanged DNA strands that are tolerated during the process of homologous recombination). Alternatively, phenotypic expression in these cells can occur if the transformed locus gets replicated, resulting in two homo-duplexes (transformed and original allele), and then transcribed before division of the cell (*Dalia and Dalia, 2019*).

## Single-cell quantification of homeologous recombination highlights transformation bottlenecks

The constructed system now allows us to quantify successful homeologous recombination events at the single-cell level, without the bias introduced by traditional plating assays. Previously, it was shown that the concentration of donor DNA as well as the length of the homology regions strongly influences transformation efficiency (*Keller et al., 2019*; *Lee et al., 1998*). To examine

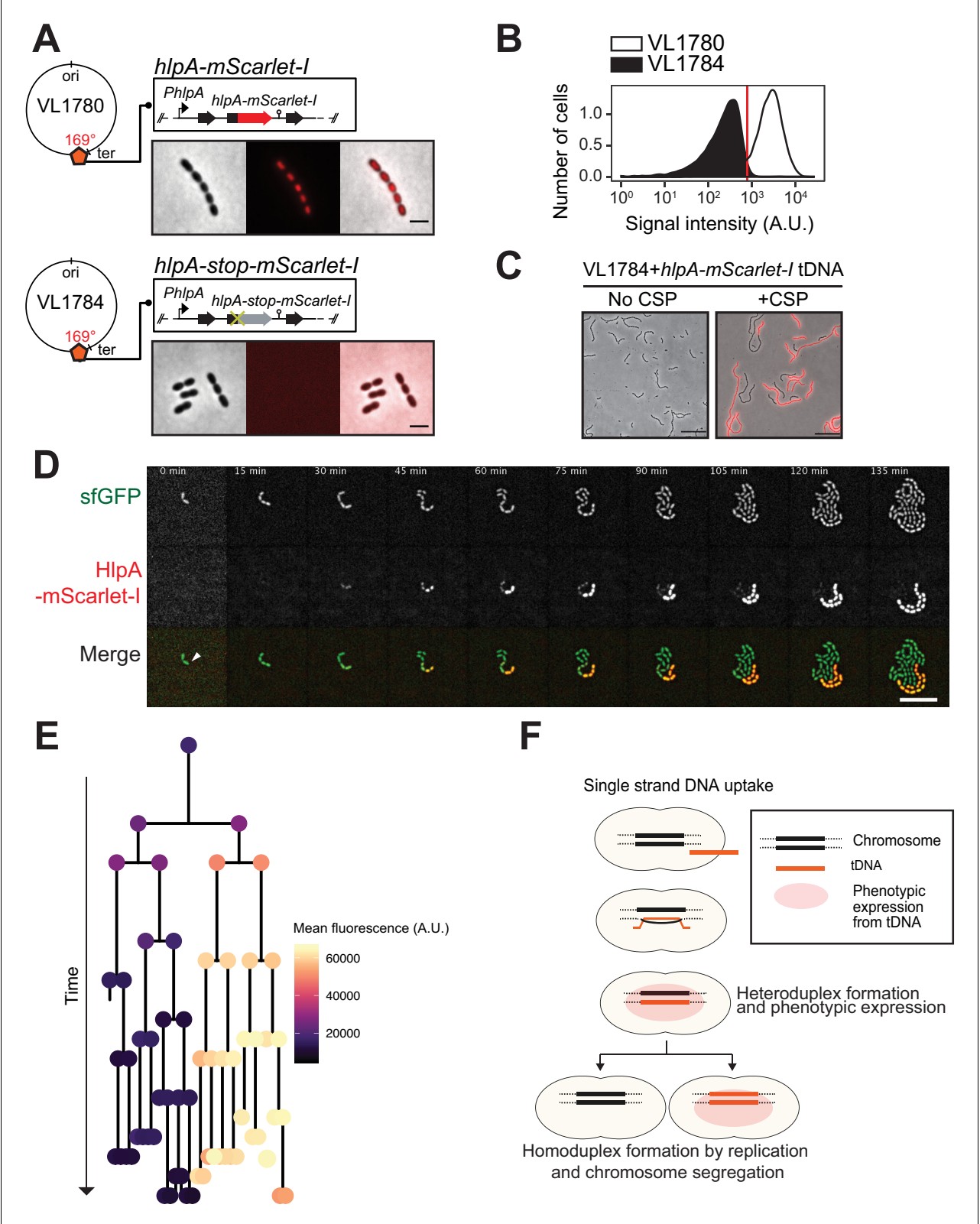

**Figure 3.** Development of a fluorescence-based real-time reporter for genetic transformation in *S. pneumoniae*. (**A**) Schematic representation of the reporter system. *hlpA-mScarlet-I* was inserted downstream of the native *hlpA* locus as a second copy (strain VL1780), resulting in red-fluorescently marked nucleoids as shown by fluorescence microscopy (Left: phase contrast, middle: red fluorescence, right: overlay, scale bar: 4 μm). A single nucleotide mutation generating a stop codon was introduced in the linker sequence between *hlpA* and *mScarlet-I*, resulting in non-fluorescent strain

*Figure 3 continued on next page*

Figure 3 continued

VL1784 (*hlpA-stop-mScarlet-I*). (**B**) Flow-cytometry measurement of *hlpA-mScarlet-I* signal of VL1780 (*hlpA-mScarlet-I*, white) and VL1784 (*hlpA-stop-mScarlet-I*, gray). (**C**) CSP-treated (right) or untreated (left) VL1784 was provided with tDNA (*hlpA-mScarlet-I*) and analyzed by fluorescence microscopy after 4 hr of incubation. Scale bar, 20 µm. (**D**) Time-lapse visualization of transformation with *hlpA-mScarlet-I* in VL1832 (VL1784+constitutively expressing cytoplasmic sfGFP). VL1832 was treated with CSP for 10 min, tDNA added (*hlpA-mScarlet-I*) for 10 min, and then spotted on C+Y agarose pad to start time-lapse imaging with a 5-min interval. Signal of constitutively expressed cytoplasmic sfGFP (top, green in the overlay) was used for cell segmentation in image analysis. Successfully transformed cells were detected by expression of HlpA-mScarlet-I (middle, red in the overlay). Scale bar, 10 µm. Also see *Video 1*. (**E**) Cell lineage tree with superimposed fluorescence intensity was built based on the time-lapse image shown in E. The quantified mean mScarlet-I signal intensity of each cell during its cell cycle was plotted as a color-coded dot onto the lineage tree with each dot corresponding to the moment of 'birth'. Note that the tree represents the lineage from only one of the two progenitor cells (indicated by white arrow in panel d). (**F**) Working model for DNA integration and chromosomal segregation of the transformed allele. tDNA is internalized as ssDNA, and recombines to replace one strand on the host chromosome forming a hetero-duplex after recombination. Following chromosomal replication and segregation, the two daughter cells have distinct homo-duplexes (either the original allele or the tDNA allele). Note that initial hetero-duplex formation might permit for phenotypic expression from the newly acquired allele if the noncoding strand is replaced by tDNA (see main text).

The online version of this article includes the following figure supplement(s) for figure 3:

**Figure supplement 1.** Classical methodology for transformation frequency estimation by antibiotics selection.
**Figure supplement 2.** Fluorescence-based detection of successful genetic transformation.
**Figure supplement 3.** Growth curves of the reporter strains.
**Figure supplement 4.** Effect of multicellular filament disruption by beadbeater on estimation of transformation frequency in flow-cytometry analysis.

recombination bottlenecks in our single-cell setup, we treated our reporter recipient strain VL1784 with CSP and used intact *hlpA-mScarlet-I* donor tDNA with various lengths of homology surrounding the stop codon (fragments of 2.7, 5, or 7 kb) at a range of different concentrations (0.0032, 0.032, 0.32, or 3.2 nM). Then, after 4 hr incubation in liquid medium to allow for complete homo-duplex allele formation and dilution of nongenetically inherited HlpA-mScarlet-I, cells were separated from chains by vigorously shaking on a bead beater devise (see *Figure 3—figure supplement 4*). Finally, transformation efficiencies were quantified by flow-cytometry (*Figure 5A*). In line with studies using classical plating methods to assess transformation efficiencies (*Keller et al., 2019*; *Lee et al., 1998*), higher transformation frequencies were observed at higher donor DNA concentrations and with longer homology regions (*Figure 5B*). Interestingly, the frequency of transformation plateaued at ~50% regardless of the concentration of donor DNA and sequence homology length (*Figure 5B*). This is in contrast to reported transformation frequencies using traditional plating assays where transformation frequencies of higher than 75% (*Ephrussi-Taylor, 1958*) and up to 100% (*Marie et al., 2017*) have been reported. This discrepancy can be explained by the lack of separation of transformed from non-transformed cells within the counted colony (*Figure 3—figure supplement 1*). To exclude the possibility that the observed limitation in transformation frequency is due to an unique feature of the *hlpA-stop-mScarlet-I* reporter, we constructed an alternative reporter cassette in which we translationally fused the superfolder green fluorescent protein (sfGFP) and SPV_1159, a nonessential small membrane protein under control of the strong constitutive P3 promoter (*Keller et al., 2019*; *Sorg et al., 2015*) cloned into the transcriptionally silent CEP locus at 295° on the circular chromosome (*Figure 5C*, strain VL1786). Based on this construct, a recipient strain was constructed containing a stop codon mutation in the linker between *spv_1159* and *sfGFP* (strain VL1788). Indeed, this *spv_1159-sfGFP*-based transformation reporter demonstrated similar transformation characteristics as the *hlpA-mScarlet-I* reporter in time-lapse microscopy and flow-cytometry analysis (*Figure 5—figure supplement 1* and *Video 3*). The transformation frequency of the *spv_1159-sfGFP* reporter was also dependent on donor DNA concentration and never exceeded ~50% (*Figure 5B*).

These data show that there is a limit on the maximum efficiency of transformation, despite the fact that most cells become competent and bind extracellular DNA (*Figure 2*) and support a model in which in general only one of the recipient allele strands is replaced by the donor DNA (*Ephrussi-Taylor, 1966*; *Figure 3F*). Importantly, these experiments indicate that during competence-dependent transformation, given the donor DNA is of sufficient (homology) length and concentration (see Discussion), in principle all targeted loci can be replaced at least on one strand.

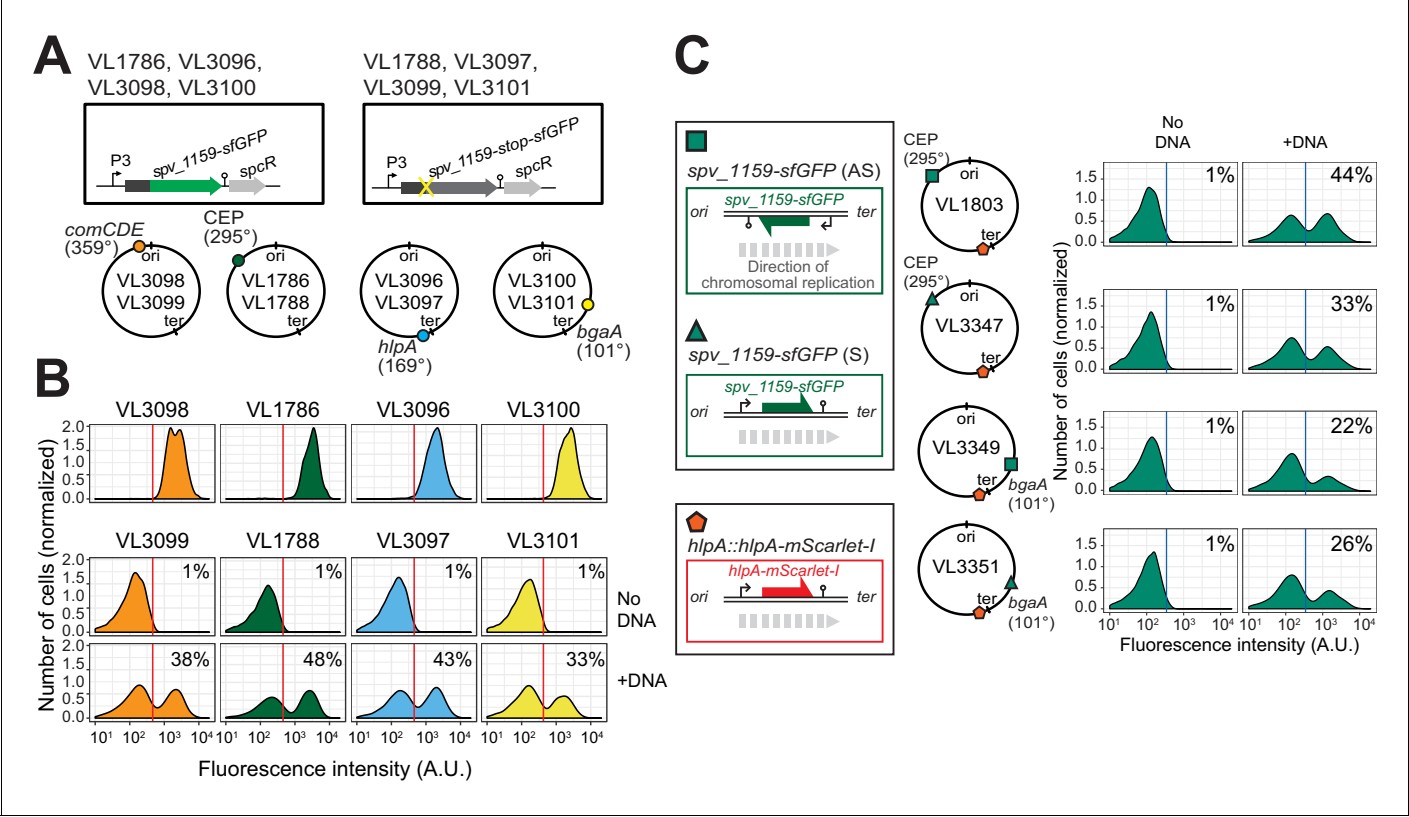

**Figure 4.** Effect of chromosomal position and strand on recombination potential. (**A**) The spv_1159-sfGFP reporter was cloned into various loci: CEP (295°; VL1786, VL1788), *hlpA* (169°; VL3096, VL3097), *comCDE* (359°; VL3098, VL3099) or *bgaA* (101°; VL3100, VL3101). A point mutation that generates a stop codon was introduced in the linker sequence between *spv_1159* and *sfGFP* for each strain (VL1788, VL3097, VL3099 or VL3101). (**B**) Flow-cytometry analysis on transformations with intact *spv_1159-sfGFP* tDNA. Strain VL1788, VL3097, VL3099 or VL3101 was transformed with intact *spv_1159-sfGFP* tDNA with 5 kb homology arm at the final concentration of 3.2 nM. 4 hr post transformation, cells were separated by beat beating and analyzed by flow-cytometry. The red vertical line indicates the threshold of positive cells in spv_1159-sfGFP signal expression. (**C**) Genetic orientation effect on transformation efficiencies. All dual reporter strains VL1803, VL3347, VL3349, or VL3351 harbor both *hlpA-stop-mScarlet-I* and *spv_1159-stop-sfGFP* reporters, but *spv_1159-sfGFP* was cloned at distinct chromosomal positions and different reading directions. *spv_1159-stop-sfGFP* was cloned at the CEP (295°) or *bgaA* (101°) locus resulting in strains VL1803/VL3347 or VL3349/3351, respectively. The coding strand of *spv_1159-sfGFP* was cloned either in the same direction as the DNA replication fork (read direction on the leading strand) (green triangle, strains VL3347/3351) or in the opposite direction (read direction on the lagging strand) (green square, strains VL1803/3349). The strains were treated with CSP and transformed with corresponding *spv_1159-sfGFP* tDNA (5 kb, 3.2 nM) alone. Transformants were analyzed by flow-cytometry. Vertical blue lines represent the threshold for green fluorescence intensity. Experiments were performed at least three times and FACS analysis of a typical experiment is shown.

The online version of this article includes the following figure supplement(s) for figure 4:

**Figure supplement 1.** Chromosomal position effect on correlation between transformation frequency and tDNA concentration.

**Figure supplement 2.** Effect of chromosomal strand to be replaced by tDNA on progression of phenotypic expression during cell division.

## Recombination with tDNA is RecA-dependent and independent of mismatch repair

Previous work showed that pneumococcal genetic transformation involves the DNA mismatch repair (MMR) system, which is mediated by HexA (*Claverys and Lacks, 1986*), and it was suggested that certain alleles upon transformation might be particularly prone to repair (*Ephrussi-Taylor, 1966*). To test whether *hexA* plays a role in our reporter system, we quantified transformation efficiencies in a *hexA* mutant background. This showed no significant recombination differences compared to the wild-type background (*Figure 4—figure supplement 2*). To test if our transformation reporter system depends on the competence-induced homologous recombinase, RecA, we depleted RecA expression level using CRISPR interference (*Liu et al., 2017*; *Figure 5D*). In control strain VL3485 (P_lac_*dcas9*, without sgRNA), induction of dCas9 by IPTG did not affect the transformation efficiency with *hlpA-mScarlet-I* tDNA. However, when RecA expression was depleted by induction with IPTG in

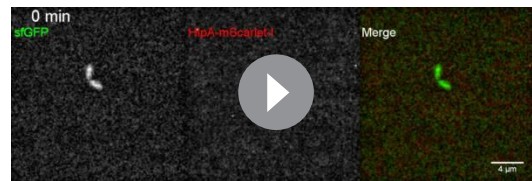

**Video 1.** Visualization of transformation using *hlpA-mScarlet-I* reporter. Shown is a movie of transformation with *hlpA-mScarlet-I* tDNA fragment in VL1832 (*hlpA-stop-mScarlet-I*) depicted in *Figure 3E*. Images are sfGFP (left), mScarlet-I (middle) and merged image (right). Frame interval, 5 min. Scale bar, 4 μm.
https://elifesciences.org/articles/58771#video1

strain VL3486 (P$_{lac}$_*dcas9*+sgRNA-recA), the transformation frequency was decreased in an IPTG-dose dependent manner. Note that although RecA is known to be critical for optimal growth in *S. pneumoniae* (*Mortier-Barrière et al., 1998*), the CRISPRi depletion levels during competence did not affect bacterial growth (*Figure 5—figure supplement 3*). Collectively, these data show that our fluorescence-based transformation assay faithfully reflects RecA-dependent homeologous recombination events.

## Cell-cycle independent homeologous recombination

It was previously suggested that the genomic location and the cell-cycle state might influence transformation efficiency as each hetero-duplex needs to be resolved to a homo-duplex by DNA replication and loci located close to *oriC* will have multiple copies (*Bergé et al., 2013*; *Dalia and Dalia, 2019*; *Ephrussi-Taylor and Gray, 1966*; *Porter and Guild, 1969*). To test whether the genomic location of the recombination site and the read orientation of the newly acquired functional allele influences transformation efficiency, we introduced the *spv_1159-stop-sfGFP* reporter at four different positions on the chromosome: on the right arm of the chromosome at 101° of the circular chromosome (*bgaA* locus), near the terminus at 169° (*hlpA* locus), on the left arm of the chromosome at 295° (*cep* locus) and near *oriC* at 359° (*comCDE* locus) (*Figure 4A and B*). In addition, *spv_1159-stop-sfGFP* was introduced on both the positive and negative strand on the left arm of the chromosome (*cep* locus at 295°) and on the right arm of the chromosome (*bgaA* locus at 101°) (*Figure 4C*). As shown in *Figure 4*, and *Figure 4—figure supplement 1*, transformation efficiencies were of a similar order across all tested loci and genetic orientations, with a maximal recombination efficiency of approximately 50%. We do note that certain loci consistently demonstrate higher transformation efficiencies than others (cf. CEP locus vs *bgaA* locus, *Figure 4—figure supplement 1*), but no significant differences were observed regarding the orientation of the construct (see Discussion).

By performing time-lapse microscopy and tracking cell fates across several generations, we can, in principle, tell whether there was a preference for integration at either the leading or lagging strand (*Figure 4—figure supplement 2A*). By placing the direction of transcription of the reporter on the leading strand, RNAP will thus use the noncoding strand as template. In this situation, only if the noncoding strand is replaced by the donor DNA, fluorescence will be apparent during the first cell cycle upon transformation. If the donor DNA is integrated at the coding strand, it will take one more replication cycle before the hetero-duplex is resolved and the noncoding strand contains the intact reporter and fluorescence will be observed later than in the first case (*Figure 4—figure supplement 2B*). Indeed, we can observe all different scenarios with transformants rapidly expressing HlpA-mScarlet-I (possible noncoding strand or double stranded recombinants) and cells that

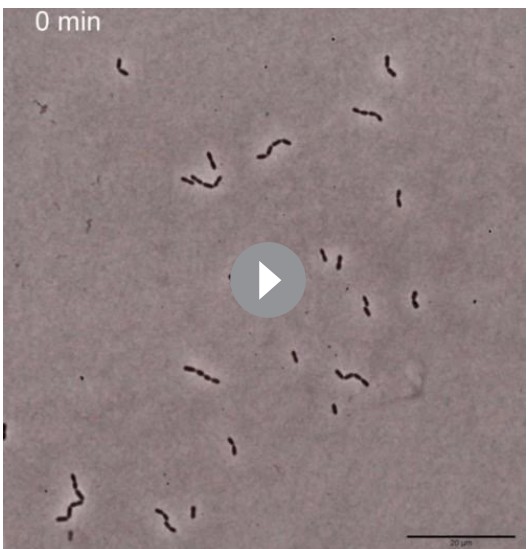

**Video 2.** Visualization of population dynamics after transformation using *hlpA-mScarlet-I* reporter. Shown is a movie of transformation with *hlpA-mScarlet-I* tDNA fragment in VL1832 (*hlpA-stop-mScarlet-I*) in lower magnification. Images are merged of phase contrast and mScarlet-I fluorescence signal. Frame interval, 5 min. Scale bar, 20 μm.
https://elifesciences.org/articles/58771#video2

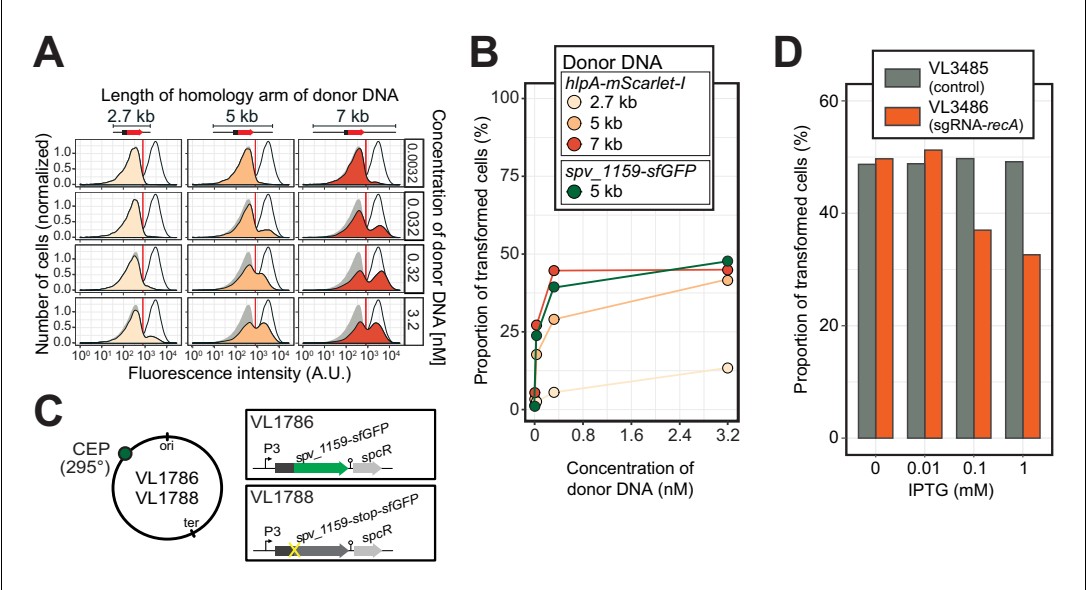

**Figure 5.** Single-cell quantification of recombination reveals an upper level of transformation efficiency. (**A**) Quantification of transformation frequency by flow-cytometry. CSP-treated VL1784 was transformed with *hlpA-mScarlet-I* tDNA of various lengths (2.7, 5, or 7 kb) at differing tDNA concentrations (0.0032, 0.032, 0.32, or 3.2 nM). The single nucleotide variant to repair the point mutation is located in the middle of each fragment (*Figure 3—figure supplement 2*). After 4 hr incubation post tDNA addition, cell chains were separated (see *Figure 3—figure supplement 4*) and analyzed by flow-cytometry. Negative control without any donor DNA (filled gray histogram) and positive control (VL1780, open histogram) is shown in all panels. Red vertical line indicates the threshold used to score mScarlet-I positive cells. (**B**) Correlation between transformation frequency and donor DNA concentration. Transformation frequency was plotted against final concentration of donor DNA. Frequency was calculated by dividing the number of cells with a FL intensity above the threshold by the total number of cells based on flow-cytometry data, as shown in panel A. (**C**) Alternative transformation reporter present on a different chromosomal position. The *spv_1159-sfGFP* reporter fusion was cloned into the CEP locus (295°; VL1786, VL1788). A point mutation resulting in a stop codon was introduced in the linker sequence separating *spv_1159* and *sfGFP* in VL1788. (**D**) Reduction of transformation efficiency by CRISPRi-based *recA* depletion. CRISPRi-based depletion strains VL3485 (P*lac_dcas9*, no sgRNA control) and VL3486 (P*lac_dcas9*, sgRNA targeting *recA*) were introduced in the *hlpA-stop-mScarlet-I* reporter strain. Strains were pre-grown with or without IPTG (0, 0.01, 0.1 or 1 mM) in acidic C+Y (pH 6.8), and then incubated with CSP (100 ng/μL) in fresh C+Y (pH 7.8) provided with donor tDNA (5 kb length, 0.32 nM). After 4 hr of phenotypic expression, transformed cells were analyzed by flow-cytometry.

The online version of this article includes the following figure supplement(s) for figure 5:

**Figure supplement 1.** Time-lapse imaging of transformation with *spv_1159-sfGFP* fragment in VL1788.

**Figure supplement 2.** Effect of *hexA* knockout on transformation frequency in the fluorescence-based reporter.

**Figure supplement 3.** Effect of *recA* knock-down on bacterial growth.

only express HlpA-mScarlet-I after the first cell division (possible coding strand recombinants) (*Figure 4—figure supplement 2C*, *Video 4*). As we did not simultaneously track DNA replication in these cells, we cannot exclude the possibility that after transformation, a round of replication occurs before phenotypic expression. Nevertheless, together with the 'bulk' (FACS) single-cell transformation experiments described above, the time-lapse data strongly suggest that there is no preference for integration at either the leading or lagging strand and that this is an unbiased event. These findings correspond with work done in the 1960s and 1970s that showed that either strand of the incoming dsDNA is degraded randomly by EndA and either strand has a similar chance of being integrated (*Puyet et al., 1990*). Recent work in *V. cholerae* demonstrated that 7% of transformation events occurred at both strands of the integration site, and it was speculated that this was because of integration of multiple donor ssDNA's replacing both the leading and lagging strand of the recipient. By recording 76 single-cell transformation events using time-lapse microscopy, we found six cases in which both daughter cells (7.8%) expressed fluorescence, suggestive of double transformation events on both strands. These findings also indirectly indicate that hetero-duplex DNA can be transcribed by RNAP and do not necessarily require a round of DNA replication to form homo-duplex DNA (see below and [*Uptain and Chamberlin, 1997*]). Together, these data show that

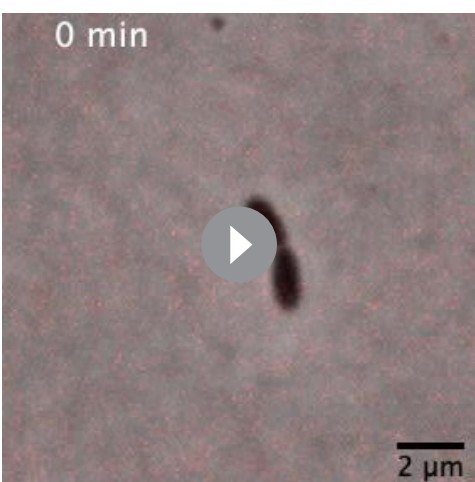

**Video 3.** Visualization of transformation using *spv_1159-sfGFP* reporter. Shown is a movie of transformation with *spv_1159-sfGFP* tDNA fragment in VL1788 (*spv_1159-stop-sfGFP*) depicted in **Figure 4—figure supplement 1**. Images are phase contrast (left), sfGFP (middle), merged image (right) Frame interval, 5 min. Scale bar, 4 μm.

https://elifesciences.org/articles/58771#video3

**Video 4.** Visualization of differential phenotypic expression timing on sense strand and anti-sense strand transformation. Shown is a movie of transformation with *hlpA-mScarlet-I* tDNA fragment in VL1832 (*hlpA-stop-mScarlet-I*) depicted in **Figure 4—figure supplement 2**. Images are an overlay between phase contrast and mScarlet-I signal. White arrow indicates a likely anti-sense strand transformed cell, and the yellow arrow indicates a likely sense strand transformed cell (See text for details). Frame interval, 5 min. Scale bar, 4 μm.

https://elifesciences.org/articles/58771#video4

hetero-duplexes with exogenous DNA are made across all available loci regardless of reading strand or distance to *oriC*.

## Direct observations of multiple recombination events in single recipients

The previous experiments demonstrated that, under ideal conditions with long flanking homology regions and high DNA concentrations, all available recombination sites are transformed on at least one of the strands. Previous studies demonstrated that pneumococcal natural transformation is capable to deal with multiple donor DNAs for genetic recombination (**Dalia et al., 2014**; **Lam et al., 2020**). Also, it has been reported that the DNA uptake and recombination process in *S. pneumoniae* is complete within 15 min (**Bergé et al., 2003**), which is a shorter time window than the doubling time (**Ephrussi-Taylor, 1966**). In order to investigate the possibility of visualizing multiple recombination events, we constructed a dual reporter strain (strain VL1803), which harbors both *hlpA-stop-mScarlet-I* and *spv_1159-stop-sfGFP* at distinct chromosomal locations (**Figure 6A**). Transformation efficiencies of this reporter strain with each single donor DNA at the saturated concentration typically reached 50% for both *hlpA-mScarlet-I* and *spv_1159-sfGFP* as quantified by microscopy (**Figure 6C–D**). When both donor DNAs were provided, double transformants were observed (15.6 ± 4.4%) as well as single *hlpA-mScarlet-I* transformants (20.2 ± 8.9%) and single *spv_1159-sfGFP* transformants (15.2 ± 4.9%). Time-lapse imaging of competent recipient VL1803 cells with both donor DNAs clearly demonstrated that single recipients could successfully recombine both fragments (**Figure 6B and C**, **Videos 5,6**). We note that, on average, the fraction of non-transformed cells is close to 50% (48.9 ± 9.5%), implying that each recombination event is not independent from the next or that there is an upper limit to the number of successful recombinations, otherwise we would expect the fraction of non-transformed cells to decrease with multiple donor DNAs (**Figure 6—figure supplement 1**). An alternative model could be that each recombination event is independent from the next but due to recombination events outside the stop codon SNP, which cannot be quantified in our setup, a reduced transformation efficiency is recorded (see Discussion).

To further explore whether transformation efficiency indeed has a plateau, we constructed a triple reporter strain (VL3127) that harbors *ftsZ-stop-mTurquoise2*, *spv_1159-stop-msfYFP*, and *hlpA-stop-mScarlet-I* at three different genomic locations (**Figure 7A**). Beside the fact that the fluorescent proteins used are spectrally distinct, every fluorescent reporter also has a specific cellular localization, facilitating automated image analyses of successful recombination. The triple reporter strain was transformed with donor tDNA fragments *ftsZ-mTurquoise2*, *spv_1159-msfYFP*, and *hlpA-mScarlet-I*.

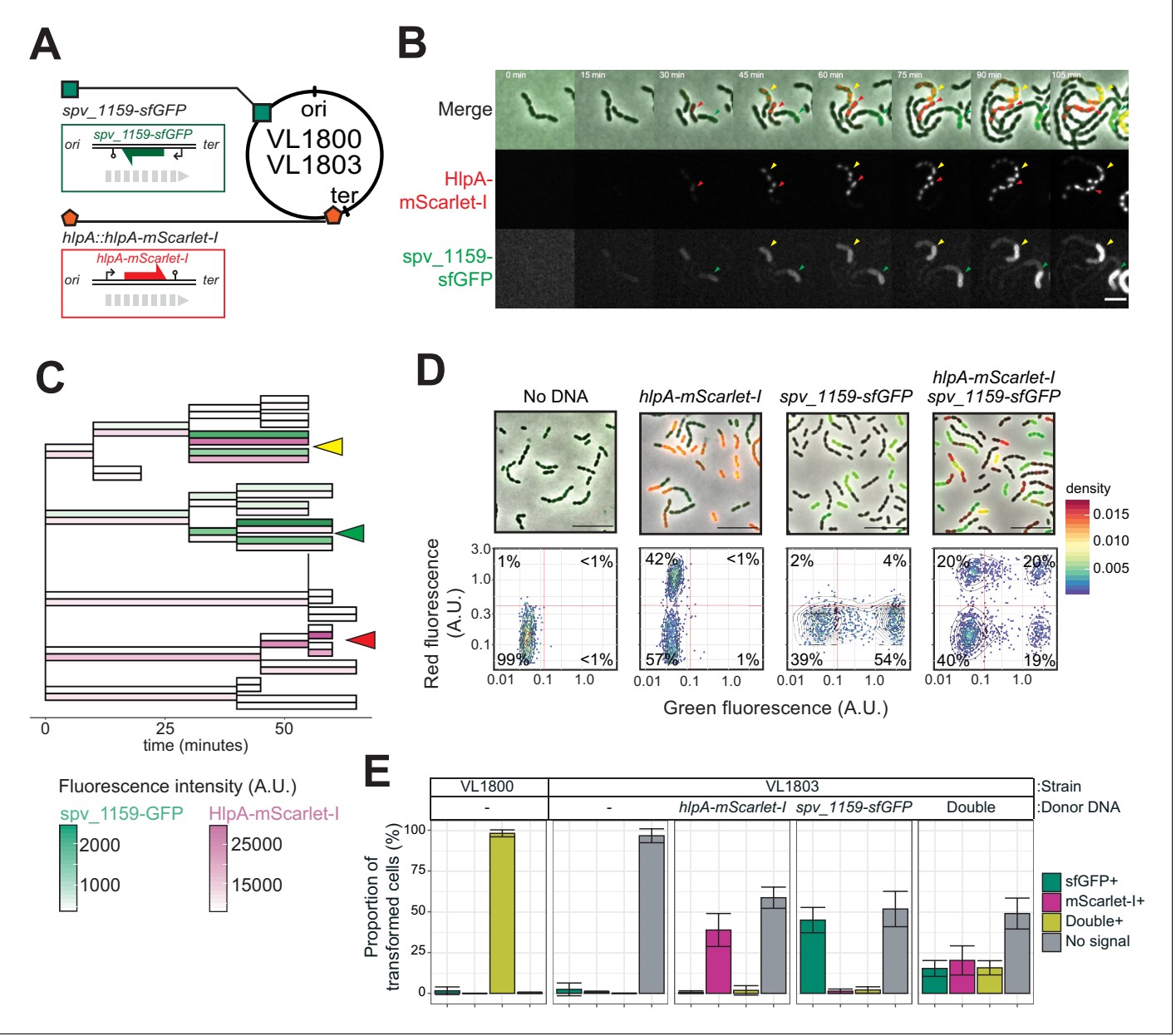

**Figure 6.** Dual transformation at distinct chromosomal positions. (**A**) Graphical representation of dual reporter strains. Dual transformation reporter strain VL1804 harbors the two transformation reporter constructs *hlpA-stop-mScarlet-I* and *spv_1159-stop-sfGFP*, at the *hlpA* and *CEP* loci, respectively. Donor tDNA (5 kb) *hlpA-mScarlet-I* and *spv_1159-sfGFP* were amplified from strain VL1800 and used for transformation. (**B**) Time-lapse visualization of double transformation. Dual reporter strain VL1803 (*hlpA-stop-mScarlet-I, spv_1159-stop-sfGFP*) was treated with CSP for 10 min, provided with 3.2 nM of both *hlpA-mScarlet-I* and *spv_1159-sfGFP* tDNAs (5 kb) for 10 min, and then spotted on a C+Y agarose pad to start time-lapse imaging at 5 min intervals. Successfully transformed cells were detected by expression of HlpA-mScarlet-I (middle, red in merge) and spv_1159-sfGFP (bottom panels, green in merge). Red and Green arrows indicate single transformed cells with *hlpA-mScarlet-I* and *spv_1159-sfGFP* tDNA, respectively. Yellow arrows indicate doubly transformed cells. Scale bar: 4 μm. See *Video 5*. (**C**) Cell lineage tree with superimposed fluorescence intensity was built based on the time-lapse image shown in B. Means of mScarlet-I (pink) and sfGFP (green) signal intensity of each cell was calculated and displayed with a color bar. Yellow, green, and red arrows indicate double transformed, single *spv_1159-sfGFP*-transformed, and single *hlpA-mScarlet-I*-transformed lineages, respectively. (**D**) Snap shots and quantitative image analysis of transformed populations. Strain VL1803 was transformed with single (*hlpA-mScarlet-I* or *spv_1159-sfGFP*) or double (*hlpA-mScarlet-I*/*spv_1159-sfGFP*) tDNA(s) (5 kb) at final concentration of 3.2 nM. After 4 hr of incubation, still images were obtained, and the fluorescence intensities were quantified and plotted. Scale bar: 10 μm. Experiments were performed at least three times and microscopy analysis of a typical experiment are shown.( **E**) Proportion of transformed phenotypes. Stacked bars represent the fraction of single transformed (red or green), double transformed (yellow), and non-transformed (gray) cells. Population of each transformed phenotype was quantified from microscopy images. Bars represent mean ± SD of three independent replicates. Analyzed data of the positive control strain VL1800 (*hlpA-*

*Figure 6 continued on next page*

*Figure 6 continued*

mScarlet-I, *spv_1159-sfGFP*) and negative control non-transformed strain VL1803 (*hlpA-stop-mScarlet-I*, *spv_1159-stop-sfGFP*) are also shown, demonstrating the accuracy of the threshold used to score positive transformants.

The online version of this article includes the following figure supplement(s) for figure 6:

**Figure supplement 1.** Expected outcome of genetic proportion after saturated transformation.

After 4 hr of incubation for fluorescent protein maturation and chromosomal segregation, cells were assessed by fluorescence microscopy. As shown in *Figure 7B* and *Video 7*, multiple transformed cells with double or triple acquired fluorescence signals were readily observed. Next, we performed single-cell transformation assays with strain VL3127 providing one tDNA or all three tDNAs and automatically quantified recombination efficiencies using Oufti and BactMAP-based image analysis (*Paintdakhi et al., 2016*; *van Raaphorst et al., 2020*; *Figure 7C*). In line with our previous observations, each single transformation with a saturated concentration of donor tDNA resulted in a recombination efficiency not higher than 50% (*Figure 7C*). Interestingly, every possible recombination event happened within the population: cells were observed in which just a single recombination event took place (the most occurring type of transformation), two recombination events (2.2 ± 0.9%, 4.1 ± 2.7% or 2.1 ± 1.8% for each possible combination) or even three recombination events (1.5 ± 1.1% of all cells). Nevertheless, more than half of the population (58.7 ± 13.4%) did not show any fluorescence when simultaneously transformed with three tDNAs. These observations support a model in which each transformation is in principle independent from the next (*Figure 6—figure supplement 1*).

## Non-homologous DNA competes with homologous DNA to reduce transformation efficiency

The data so far show that in principle every locus in *S. pneumoniae* can be efficiently transformed to a maximum of 50% of the cells when providing tDNA at high DNA concentrations and with long homology arms. However, when providing multiple tDNAs, the untransformed fraction even increases and becomes greater than 50%. Recently, it was shown using whole genome sequencing of transformation events occurring during contact-dependent DNA uptake, that a single recipient could have at least 29 different recombination events (*Cowley et al., 2018*). Together, this suggests that many recombination events are likely going unnoticed in our single-cell transformation assay, and that these recombination events become limiting, as we can only detect successful recombination when the stop codon in our fluorescent reporter is replaced for a functional allele. If this is true,

adding non-specific DNA would compete with donor tDNA resulting in reduced transformation efficiencies. To test this, we utilized homology-unrelated *E. coli*-derived DNA fragments of 5 kb with a similar GC content to *S. pneumoniae* as competing donor DNA. Indeed, as shown in *Figure 7E*, co-transformation of strain VL1803 (*hlpA-stop-mScarlet-I*, *spv_1159-stop-sfGFP*) with *E. coli* DNA significantly reduced the transformation efficiency. When 0.32 µM of *hlpA-mScarlet-I* tDNA alone was used as 7 kb donor DNA, approximately 43% of cells were transformed. However, when 0.32 µM of *hlpA-mScarlet-I* tDNA was given in the presence of saturating amounts of *E. coli* DNA (3.2 µM), only 3% of transformants were observed. Together, these data suggest that the fact that we never reach transformation efficiencies higher than 50% of the population even in the presence of multiple tDNAs, is because of saturation of the DNA uptake and integration machinery. The

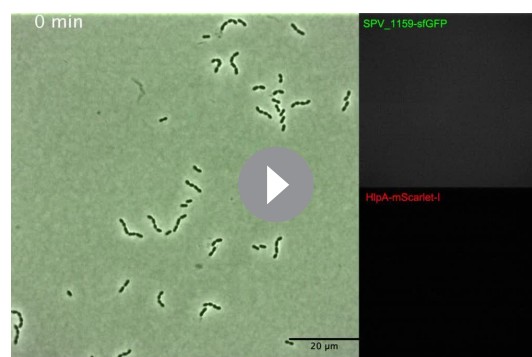

**Video 5.** Visualization of double transformation in the population. Shown is a movie of double transformation with *hlpA-mScarlet-I* and *spv_1159-sfGFP* tDNA fragments in dual reporter strain VL1803 (*hlpA-stop-mScarlet-I*, *spv_1159-stop-sfGFP*). White arrow indicates a double transformed cell lineage. Frame interval, 5 min. Scale bar, 20 µm.
https://elifesciences.org/articles/58771#video5

**Video 6.** Visualization of double transformed cells. Shown is an enlarged movie of *hlpA-mScarlet-I* and *spv_1159-sfGFP* tDNA fragments in dual reporter strain VL1803 (*hlpA-stop-mScarlet-I*, *spv_1159-stop-sfGFP*) depicted in *Figure 6B*. Images are mScarlet-I (left), sfGFP (middle), merged image (right) Frame interval, 5 min. Scale bar, 4 μm.

https://elifesciences.org/articles/58771#video6

saturation can be caused by non-successful recombination events with donor DNA or successful recombination events with the donor DNA but outside of the stop SNP that cannot be detected in the single-cell assay (*Figure 6—figure supplement 1*).

## Efficient HGT in sessile co-cultures

So far, we demonstrated that transformation is highly efficient under ideal and saturated experimental settings in which pure PCR products were used as donor DNA. Previous studies showed that natural environments also promote efficient HGT (*Cowley et al., 2018*; *Domenech et al., 2020*). Indeed, in pneumococcal biofilms, which are a model for nasopharyngeal colonization, competence and fratricide are strongly induced, resulting in high relative concentrations of DNA. To assess transformation potential under more realistic conditions, we tested transformation efficiency in a co-culture system in which two pneumococcal strains are grown together on a semi-solid surface without adding synthetic CSP (*Figure 8A*; see Materials and methods section for detail procedure). HGT in such systems is based on genomic DNA released by dead cells followed by DNA uptake and transformation of competent recipient cells (*Domenech et al., 2020*). Here, we used strain R895 (*cmR*) as recipient and strain R4692 (Δ*comCDE*, *smR*, *novR*) as donor. Both are genetically identical unencapsulated R800 derivatives (*Lefevre et al., 1979*) except for a single SNP conferring streptomycin (SNP in *rpsL*) or novobiocin (SNP in *gyrB*) resistance and a chloramphenicol resistance cassette present in the recipient R895 strain (*Figure 8A*). Strain R4692 is also unable to activate competence due to a *comCDE* deletion so transformation can only occur in one direction from donor (R4692) to recipient (R895). R895 and R4692 were pre-cultured separately until early exponential phase and then mixed in an approximate ratio of 3:7 (see Materials and methods). The mixture was immediately spotted on agar plates followed by incubation at 37°C for 4 hr to allow spontaneous competence development and transformation between strains. Cells were collected by scraping them from the plates and separated by sonication. Serial dilutions of the resulting cell suspension were plated with 4.5 μg/mL of chloramphenicol (for the recovery of the total number of viable recipient cells) and with combinations of chloramphenicol plus streptomycin (10 μg/mL) and/or novobiocin (4 μg/mL) (for the recovery of the single or double transformed recipient cells). As shown in *Figure 8B*, also in this more realistic model, very high transformation efficiencies are obtained with a single transformation efficiency with *smR* or *novR* of $5.70 \times 10^{-2}$% (SD, $5.70 \times 10^{-2}$%) or $1.75 \times 10^{-2}$% (SD, $1.68 \times 10^{-2}$%), respectively. Double transformation efficiency with both *smR* and *novR* was $8.01 \times 10^{-5}$% (SD, $9.92 \times 10^{-5}$%), which is close to the product of the single transformation efficiencies ($5.70 \times 10^{-2}$% $\times$ $1.75 \times 10^{-2}$% = $9.9 \times 10^{-4}$%). As a control, we also performed experiments using strain R4574 as donor (same genotype as R4692, but not harboring *smR* or *novR* allele), which never generated streptomycin nor novobiocin-resistant R895, demonstrating that de novo mutations conferring resistance do not occur in this experimental setup. Together, these experiments support our single- cell observations that multiple transformation events occur efficiently and independently even in more realistic settings within sessile co-cultures.

## Discussion

The species of *S. pneumoniae* is vastly diverse with a core genome of approximately 500–1100 orthologues genes and a pan-genome of 5000–7000 orthologs (*Hiller and Sá-Leão, 2018*). In addition, many genes are mosaic such as several genes encoding for penicillin-binding proteins in penicillin-resistant clinical strains (*Hakenbeck et al., 2012*). One of the main reasons for the high level of genome plasticity and rapidly changing population dynamics is because of the highly conserved competence-based transformation system present in nearly all pneumococcal genomes (*Croucher et al., 2016*). Indeed, rapid spread of antibiotic resistance alleles and capsule loci have

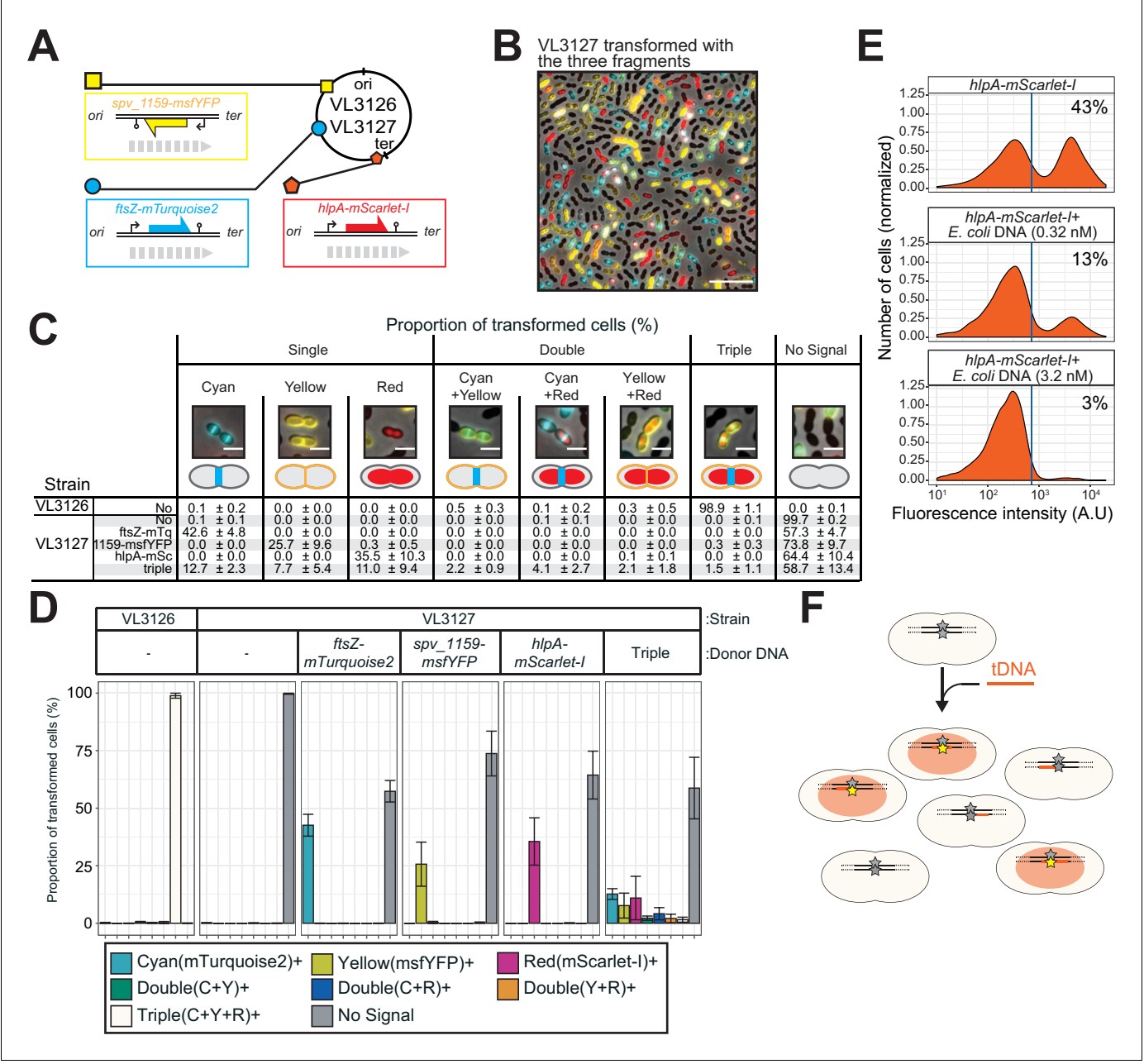

**Figure 7.** Direct observation of recombination of three separate tDNAs during a single transformation event. (**A**) Schematic representation of the triple-labeled strain VL3126 harboring three reporter cassettes: *hlpA-mScarlet-I*, *spv_1159-msfYFP*, and *ftsZ-mTurquoise2* at the *hlpA*, *CEP*, and *ftsZ* loci, respectively. Strain VL3127 contains stop codon mutations in the linker between each of the fluorescent fusion proteins. Gray arrows indicate the direction of the DNA replication fork relative to the reporter cassette. (**B**) Microscope image of strain VL3127 treated with CSP and transformed with the tDNAs of *hlpA-mScarlet-I*, *spv_1159-sfGFP*, and *ftsZ-mTurquoise2* (3.2 nM each) amplified from VL3126. Merge image of phase contrast, cyan (FtsZ-mTurquoise2), yellow (spv_1159-msYFP) and red (HlpA-mScarlet-I) fluorescence is shown. Scale bar: 10 μm. (**C and D**) Proportion of transformed phenotypes. Population of each transformed phenotype was quantified from microscope images. Representative images for each phenotype are shown. Scale bar: 2 μm. (**D**) Stacked bars represent proportion of single transformed (cyan, yellow or red), double transformed [green (cyan+yellow), blue (cyan+red), orange (yellow+red)], triple transformed (white) and non-transformed (gray) cells. Bars represent mean ± SD of three independent replicates. Analyzed data of the positive control strain VL3126 (*ftsZ-mTurquoise2*, *spv_1159-msfYFP*, *hlpA-mScarlet-I*) and negative control non-transformed strain VL3127 (*ftsZ-stop-mTurquoise2*, *spv_1159-stop-msfYFP*, *hlpA-stop-mScarlet-I*) are also shown, demonstrating the accuracy of the threshold used to score positive transformants. (**E**) Competition effect of unrelated DNA on transformation frequency. CSP-treated VL1803 was transformed with 7 kb *hlpA-mScarlet-I* tDNA at the final concentration of 0.32 nM in the absence or the presence of an unrelated DNA fragment (0.32 nM or 3.2 nM) amplified from *E. coli*. After incubation of 4 hr post transformation, cells were separated and analyzed by flow-cytometry. Red vertical line indicates the threshold of positive cells in mScarlet-I signal expression. The proportion of positive cells (%) is depicted in the plots. (**F**). Fragmented

*Figure 7 continued on next page*

*Figure 7 continued*

tDNA recombination model. The fluorescence-based reporters used in this study rely on replacement of the stop codon SNP (gray star) by intact (amino acid coding) SNP (yellow star) that is located in the middle of the tDNA fragment (orange line). All prepared tDNA molecules have obviously intact SNP, but, integration into host chromosome may take place outside the SNP, which is never distinguished from true untransformed cells by the fluorescence-based system and effectively acting as competing DNA for tDNAs that transform the SNP.

been observed among human populations under selective pressure (*Chewapreecha et al., 2014*). Here, we investigated the molecular basis for competence-dependent transformation at the single-cell level and show that the uptake, integration, and expression of tDNA are highly efficient and are largely independent from the recipient's cell-cycle stage or of the chromosomal position of the target locus. This was made possible by the setup of a sensitive real-time detection system to quantify successful homeologous recombination events. A major benefit of the here established single-cell approach over traditional plate-based assays is that it allows for the detection of more subtle effects and offers better resolution to study the kinetics of the processes involved. Indeed, using the system developed here, we could visualize and quantify the recombination of three different tDNAs in single recipient cells demonstrating the efficiency of the pneumococcal transformation process.

Genome sequencing has indicated that up to 29 recombination events may have taken place in a single round of transformation in the same cell when selecting for the transfer of an antibiotic resistance allele in *S. pneumoniae* (*Cowley et al., 2018*), while 40 recombination events have been reported in *B. subtilis* (*Carrasco et al., 2016*). Our work now provides direct evidence that this is not an anomaly and that multiple recombination events are possible during a single transformation episode, even in the absence of selection. Besides shedding light on the efficiency by which transformation can happen in *S. pneumoniae*, by imaging transformation at the single-cell level, we provide direct evidence that typically only one recipient strand is replaced during competence-dependent transformation, and that there is no bias toward replacement of the leading or lagging strand. As observed in *V. cholerae*, in approximately 7% of transformants, both strands can be replaced, which is likely caused by DNA repair leading to removal of the recipient strand on the hetero-duplex or by integration of multiple tDNAs (*Dalia and Dalia, 2019*). This is in line with predictions made using unlinked antibiotic resistance alleles (*Porter and Guild, 1969*). In addition, our single-cell observations suggest that the replaced noncoding strand by recombination within the hetero-duplex is immediately transcribed by RNAP and can lead to lineages of cells with nongenetic inherited phenotypes, or that the transformed allele is replicated and transcribed well before cell division occurs (*Figure 3*).

We show that any site regardless of its chromosomal position or orientation with regard to DNA replication can be efficiently transformed, although not with the exact same efficiencies (*Figure 4—figure supplement 1*). Possible explanations for local difference in recombination efficiency could be the levels of DNA compaction or transcription activity. As RecA-mediated DNA strand exchange is a reversible reaction in vitro (*Dutreix et al., 1991*; *Konforti and Davis, 1990*), under steady state conditions DNA strand exchange rarely reaches 50% efficiency. However, in vivo, when providing a single tDNA to competent cells, we readily reach 50% DNA strand exchange, again highlighting that this process is highly efficient under our experimental conditions.

Interestingly, we find that the percentage of untransformed cells is lower when three tDNAs are provided instead of two tDNAs (~58% vs ~49% of untransformed cells, respectively: *Figures 6* and *7*). Together with the observation

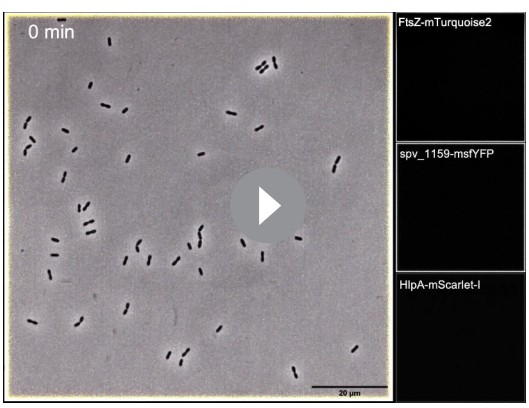

**Video 7.** Visualization of triple transformation in population. Shown is a movie of triple transformations with *ftsZ-mTurquoise2*, *spv_1159-msfYFP* and *hlpA-mScarlet-I* tDNA fragments in triple reporter strain VL1803 (*ftsZ-stop-mTurquoise2*, *spv_1159-stop-msfYFP*, *hlpA-stop-mScarlet-I*). Frame interval, 10 min. Scale bar, 20 μm.

https://elifesciences.org/articles/58771#video7

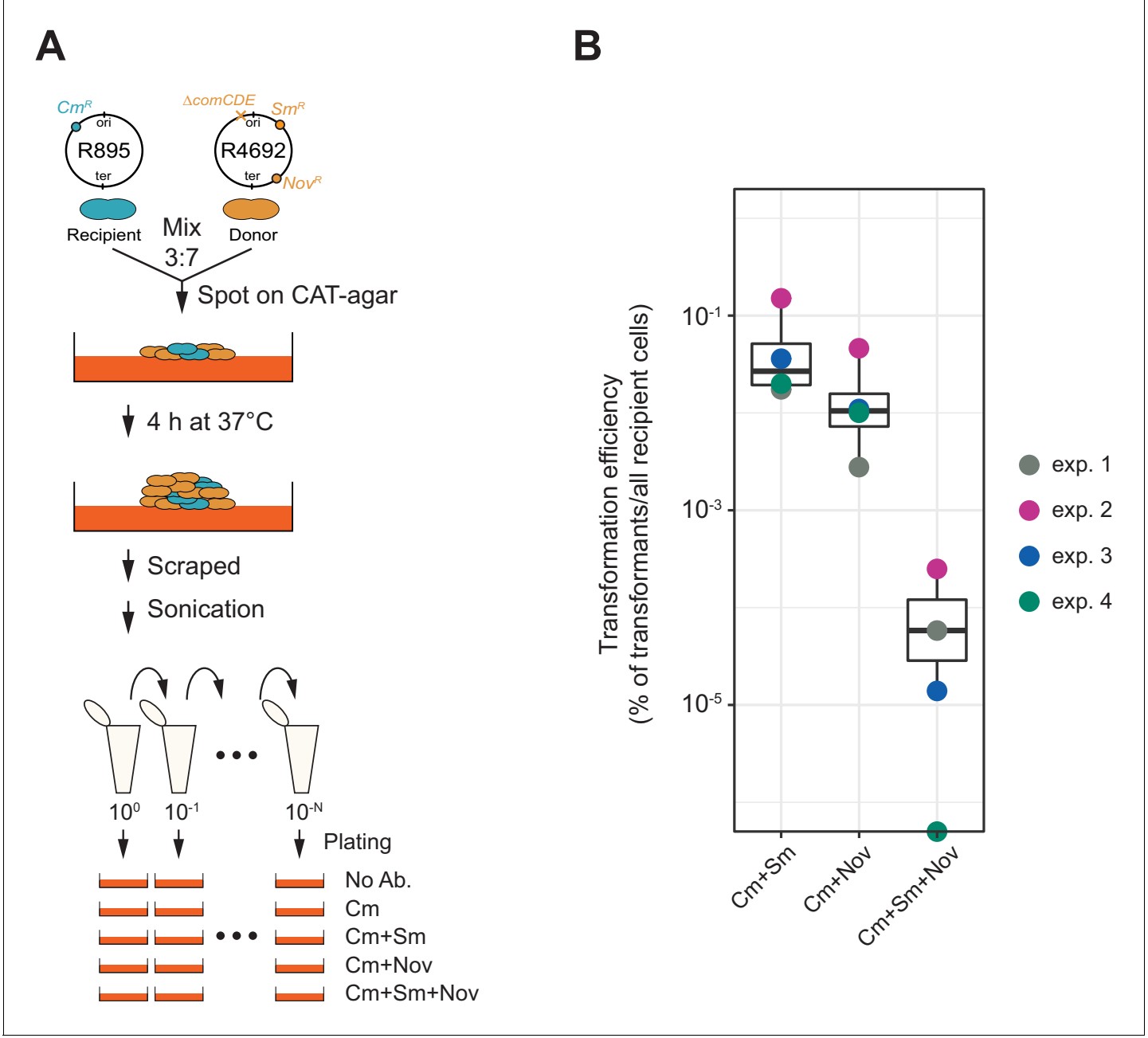

**Figure 8.** Horizontal gene transfer between *S. pneumoniae* strains. (**A**) Schematic representation of the transformation assay between *S. pneumoniae* strains. Pre-cultured recipient strain R895 (*cmR*) and donor strain R4692 (Δ*comCDE*, *smR*, *novR*) were mixed in approximately 3:7 ratio, and spotted on 3% horse blood CAT-agar (see Materials and methods). After 4 hr incubation at 37°C to allow strain-to-strain transformation, cells were scraped and separated by sonication. Then, serial dilutions of cell suspensions were plated with 4.5 µg/mL of chloramphenicol (Cm, for the recovery of the total number of viable recipient cells) and with combinations of chloramphenicol plus streptomycin (Sm, 10 µg/mL) and/or novobiocin (Nov, 4 µg/mL). (**B**) Transformation efficiency was calculated by dividing the number of transformants by the total number of viable recipient count. Four independent biological replicates were performed and box plots demonstrate the average efficiencies.

that the presence of non-homologous DNA reduced our observed transformation efficiency (*Figure 7E*), suggests that, in principle, every recombination event is independent of the next, but that many unsuccessful recombination events and successful recombination events outside the stop codon of our reporter are taking place and that this limits the efficiency of site-specific recombination (*Figure 6—figure supplement 1* and *Figure 7F*).

The overall biological implication of the limitation on competence-dependent transformation is that this mechanism ensures that in most cases one copy of the original recipient DNA remains unaltered. This might represent a fail-safe scenario so that in case a deleterious tDNA is incorporated, at least one daughter cell will survive. While this might be considered as a 'spandrel' effect: a characteristic that flows inevitably from a selected phenotype but has not been selected for directly (*Gould and Lewontin, 1979*), being able to safely sample from a large pan-genome might contribute to the vast genome plasticity and genome diversity as observed in natural pneumococcal populations. Interestingly, we also find highly efficient HGT and independent transfer of genetic markers between a donor and recipient pneumococcal strain growing together on agar plates (*Figure 8*), indicating that our single-cell observations under laboratory conditions also reflect settings that depend on lysis of the donor cell and uptake of chromosomal DNA. It will be interesting to see how efficient competence-dependent transformation and HGT is under more realistic conditions such as within polymicrobial community within a host. Future single-cell work will allow the investigation of the localization of the enzymes involved in transformation, how strand exchange during transformation occurs and what the dynamics of the molecular machines are during DNA uptake, integration, and expression of tDNA.

# Materials and methods

**Key resources table**

| Reagent type (species) or resource | Designation | Source or reference | Identifiers | Additional information |
|---|---|---|---|---|
| Strain, strain background (*Streptococcus pneumoniae*) | Various | This paper | NCBI Taxon: 1313 | See *Supplementary file 1* |
| Sequence-based reagent | Various oligonucleotides | This paper (Sigma-Aldrich) | Primers for cloning | See *Supplementary file 2* |
| Chemical compound, drug | D-Luciferine | Synchem | bc219; CAS: 115144-35-9 | |
| Software, algorithm | FIJI | doi:10.1038/nmeth.2019 | RRID:SCR_002285 | |
| Software, algorithm | Oufti | doi:10.1111/mmi.13264 | RRID:SCR_016244 | |
| Software, algorithm | BactMAP | doi:10.1111/mmi.14417 | https://github.com/veeninglab/BactMAP | |
| Software, algorithm | SuperSegger | doi:10.1111/mmi.13486 | https://github.com/wiggins-lab/SuperSegger | |

## Bacterial strains and growth condition

All pneumococcal strains used in this study are derivatives of serotype 2 *S. pneumoniae* D39V (*Avery et al., 1944*; *Slager et al., 2018*) unless specified otherwise. See *Supplementary file 1* for a list of the strains used. *S. pneumoniae* was grown in C+Y (pH 6.8) medium at 37°C. C+Y was adapted from (*Adams and Roe, 1945*) and contained the following compounds: adenosine (68.2 mM), uridine (74.6 mM), L-asparagine (302 mM), L-cysteine (84.6 mM), L-glutamine (137 mM), L-tryptophan (26.8 mM), casein hydrolysate (4.56 g L$^{-1}$), BSA (729 mg L$^{-1}$), biotin (2.24 mM), nicotinic acid (4.44 mM), pyridoxine (3.10 mM), calcium pantothenate (4.59 mM), thiamin (1.73 mM), riboflavin (0.678 mM), choline (43.7 mM), CaCl$_2$ (103 mM), K2HPO4 (44.5 mM), MgCl$_2$ (2.24 mM), FeSO$_4$ (1.64 mM), CuSO$_4$ (1.82 mM), ZnSO$_4$ (1.58 mM), MnCl$_2$ (1.29 mM), glucose (10.1 mM), sodium pyruvate (2.48 mM), saccharose (861 mM), sodium acetate (22.2 mM), and yeast extract (2.28 g L$^{-1}$).

## Strain construction

### Construction of *ssbB::ssbB_msfYFP* (VL2219)

To construct YFP reporter for *ssbB* transcription, monomeric *yfp* (*myfp*) was introduced immediately downstream of *ssbB* at the native *ssbB* locus together with an RBS. *myfp* gene was amplified with OVL1414 and OVL1417 from genomic DNA of MK308 (*parB::parB-yfp*) (*van Raaphorst et al., 2017*).

Upstream and downstream fragments were amplified with primer pairs of OVL166/OVL1196 and OVL1199/OVL167 using genomic DNA of VL599 (*ssbB::ssbB_luc,kan*R) (*Slager et al., 2014*) as template, respectively. The three resulting fragments were digested with BsmBI, ligated and transformed into *S. pneumoniae* D39V to obtain strain VL2219.

## Construction of *comGA*::*comGA-msfYFP* (VL2536)

To construct translational fusion of *comGA* and *msfYFP*, *linker-mYFP* was amplified by PCR with OVL351/OVL690 from genomic DNA of MK308 (*parB::parB-yfp*) (*van Raaphorst et al., 2017*). 'Upper' and 'downer' fragments containing *comGA* were amplified with primer pairs OVL391/OVL392 and OVL691/OVL394 using genomic DNA of D39V as template, respectively. The three resulting fragments were fused by overlap PCR and transformed into *S. pneumoniae* D39V. Transformed clones were screened by PCR and sequenced. Resulting strains were additionally transformed by *ssbB::ssbB_mScarlet-I, kanR* fragments, obtaining strain VL2536.

## Construction of *ssbB:ssbB_mScarlet-I, kanR* (VL2536)

To construct transcriptional fusion of *ssbB* and *mScarlet-I*, *mScarlet-I* was amplified by PCR with OVL1415/OVL1418 from genomic DNA of VL1787 (*cep:spv_1159-mScarlet-I, spcR*) (*Keller et al., 2019*). Upper and downer fragments were amplified with OVL166/OVL1168 and OVL1199/OVL167 using genomic DNA of VL599 (*ssbB::ssbB_luc, kanR*) as template, respectively. The three resulting fragments were fused by Golden Gate assembly using BsmBI and transformed into *S. pneumoniae* strain, *comGA::comGA-msfYFP*. Transformed clones were selected with kanamycin and sequenced, obtaining strain VL2536.

## Construction of *comEA*::*msfYFP-comEA* (VL2537)

To construct translational fusion of *comEA* and *msfYFP*, *mYFP-linker* was amplified with OVL2029/OVL2028 from genomic DNA of VL1818 (*comEC::msfYFP-comEC*) (Veening lab collection). Upper and downer fragments containing *comEA* were amplified with OVL354/OVL1664 and OVL2026/OVL357 using genomic DNA of VL870 (*comEA::mNeonGreen-comEA*) (Veening lab collection) as template, respectively. The three resulting fragments were fused by Golden Gate assembly using BsmBI and transformed into *S. pneumoniae* D39V. Transformed clones were screened by PCR and sequenced, obtaining strain VL2537.

## Construction of *comFA*::*comFA-msfYFP* (VL2538)

To construct a translational fusion of *comGA* and *msfYFP*, *linker-mYFP* gene was amplified with OVL351/OVL690 from genomic DNA of MK308 (*parB::parB-yfp*) (*van Raaphorst et al., 2017*). Upper and downer fragments containing *comFA* were amplified with OVL358/OVL521 and OVL1129/OVL361 using genomic DNA of D39V as template, respectively. The three resulting fragments were fused by overlap PCR and transformed into *S. pneumoniae* D39V. Transformed clones were screened by colony PCR and sequenced, obtaining VL2538 strain.

## Construction of *dprA*::*dprA-msfYFP*, *eryR* (VL3355)

To construct a translational fusion of *dprA* and *msfYFP*, *linker-mYFP* gene was amplified with OVL3481/OVL3482 from genomic DNA of *cep::spcR, P3_spv_1159-msfYFP* (codon-optimized) strain (Rueff AS and Veening JW, unpublished). Upper and downer fragments containing *dprA* were amplified with OVL3487/OVL3488 and OVL3489/OVL3490 using genomic DNA of D39V as template, respectively. Erythromycin resistance marker (*eryR*) was amplified OVL2549/OVL2771 using genomic DNA of *hexA::eryR* strain (Veening lab collection). The four fragments were fused by Golden Gate assembly using BsmBI and transformed into *S. pneumoniae* D39V. Transformed clones were selected by erythromycin and sequenced, obtaining strain VL3355.

## Construction of *hlpA::hlpA_hlpA-mScarlet-I* (VL1780)

To construct *hlpA-mScarlet-I*, the *hlpA-mScarlet-I* gene was introduced downstream of the original *hlpA* gene at its own locus as a second copy of *hlpA*. Upper and downer fragments were amplified by PCR with OVL43/OVL44 and OVL45/OVL46 using genomic DNA of MK119 (*hlpA::hlpA_hlpA-mKate2, cmR*) (*Beilharz et al., 2015*), respectively. *mScarlet-I* gene was amplified by PCR with

OVL55 and OVL56 using codon-optimized synthetic *mScarlet-I* gene as template (*Keller et al., 2019*). The three resulting fragments were fused by overlap PCR and transformed into *S. pneumoniae*. Transformed clone was selected by chloramphenicol, obtaining VL1780.

## Construction of *hlpA::hlpA_hlpA-stop-mScarlet-I* (VL1784, VL1832)

To disrupt translation between *hlpA* and *mScarlet-I*, on *hlpA::hlpA_hlpA-mScarlet-I, cmR* construct, single nucleotide mutation was introduced in domain breaking linker between *hlpA* and *mScarlet-I*. Upper or downer fragments were amplified by PCR with OVL43/OVL724 or OVL873/OVL46 using genomic DNA of VL1780 as template. The resulting fragments were fused by overlap PCR and transformed into *S. pneumoniae* D39V. Transformed clone was selected by chloramphenicol, obtaining VL1784.

To obtain VL1832 (*hlpA::hlpA_hlpA-mScarlet-I, cmR; CEP::sfGFP, spcR*), *CEP::P3_sfGFP, spcR* fragment was amplified by PCR with OVL37/OVL40 using genomic DNA of D-PEP33 (*CEP:: spcR, P3_sfGFP*) (*Sorg et al., 2015*). The fragment was transformed into VL1784, and transformed clone was selected by spectinomycin, obtaining VL1832.

## Construction of *CEP::spcR, P3_spv_1159-sfGFP* (VL1785, VL1800)

To construct membrane localizing sfGFP, hypothetical protein with transmembrane domain, spv_1159, was translationally fused to sfGFP under the control of synthetic constitutive promoter P3 at the CEP locus of the *S. pneumoniae* chromosome. Upper and downer fragments were amplified by PCR with OVL37/OVL631 and OVL634/OVL40 using genomic DNA of VL430 (*CEP:: spcR, P3_spv_1159-sfGFP*) (*Keller et al., 2019*), respectively. *spv_1159* was amplified by PCR with OVL632 and OVL633 using genomic DNA of D39V. The three resulting fragments were fused by Golden Gate assembly with BsmBI and transformed into *S. pneumoniae* D39V. Transformants were selected on Colombia agar plates containing spectinomycin, obtaining strain VL1785.

To obtain a dual-labeled strain, the *CEP::spcR, P3_spv_1159-sfGFP* fragment was amplified by PCR with OVL37/OVL40 using genomic DNA of VL1785, and transformed into VL1780 to obtain VL1800 (*hlpA::hlpA_hlpA-mScarlet-I,cmR; CEP::spcR, P3_spv_1159-sfGFP*).

## Construction of *CEP::spcR, P3_spv_1159-stop-sfGFP* (VL1788, VL1803, VL1930)

To disrupt translation between *spv_1159* and *sfGFP*, on the *CEP::spcR, P3_spv_1159-sfGFP* construct, a single nucleotide mutation was introduced in domain breaking linker between *spv_1159* and *sfGFP*. Upper and downer fragments were amplified by PCR with OVL37/OVL724 or OVL723/ OVL40 using genomic DNA of VL1785 as template. The resulting fragments were fused by overlap PCR and transformed into *S. pneumoniae* D39V. Transformants were selected by chloramphenicol, obtaining strain VL1786.

To obtain dual reporter strain VL1800, the *CEP::spcR, P3_spv_1159-stop-sfGFP* fragment was amplified by PCR with OVL37/40 using genomic DNA of VL1786, and transformed into VL1784 to obtain VL1800 (*hlpA::hlpA_hlpA-stop-mScarlet-I, cmR; CEP::spcR, P3_spv_1159-stop-sfGFP*).

## Construction of *hlpA::spcR, P3_spv_1159-sfGFP* (VL3096)

To insert the *spcR, P3_spv_1159-sfGFP* construct downstream of the *hlpA* locus, *spcR, P3_spv_1159-sfGFP* was amplified by PCR with OVL2855/OVL2856 using genomic DNA of VL1785 (*CEP:: spcR, P3_spv_1159-sfGFP*) as template. Upper and downer fragments were amplified by PCR with OVL2868/OVL2859 and OVL2860/OVL2869 using genomic DNA of D39V, respectively. The three resulting fragments were fused by Golden Gate assembly with BsmBI and transformed into *S. pneumoniae* D39V. Transformants were selected by spectinomycin, obtaining strain VL3096.

## Construction of *hlpA::spcR, P3_spv_1159-stop-sfGFP* (VL3097)

To disrupt translation between *spv_1159* and *sfGFP*, on *hlpA::spcR, P3_spv_1159-sfGFP* construct, single nucleotide mutation was introduced in domain breaking linker between *spv_1159* and *sfGFP*. Upper or downer fragments were amplified by PCR with OVL2868/OVL724 or OVL723/OVL2869 using genomic DNA of VL3096 as a template. The resulting fragments were fused by overlap PCR

and transformed into *S. pneumoniae* D39V. Transformants were selected on Colombia agar plates containing chloramphenicol, obtaining VL3097.

## Construction of *comCDE::spcR, P3_spv_1159-sfGFP* (VL3098)

To insert the *spcR, P3_spv_1159-sfGFP* construct right downstream of the *comCDE* locus, *spcR, P3_spv_1159-sfGFP* was amplified by PCR with OVL2855/OVL2856 using genomic DNA of VL1785 (*CEP::spcR,P3_spv_1159-sfGFP*) as template. Upper and downer fragments were amplified by PCR with OVL371/OVL2861 and OVL2862/OVL2870 using genomic DNA of D39V, respectively. The three resulting fragments were fused by Golden Gate assembly with BsmBI and transformed into *S. pneumoniae* D39V. Transformed clone was selected by spectinomycin, obtaining VL3098.

## Construction of *comCDE::spcR, P3_spv_1159-stop-sfGFP* (VL3099)

To disrupt translation between *spv_1159* and *sfGFP*, on *comCDE::spcR, P3_spv_1159-sfGFP* construct, single nucleotide mutation was introduced in domain breaking linker between *spv_1159* and *sfGFP*. Upper or downer fragments were amplified by PCR with primers OVL371/OVL724 or OVL723/OVL2870 using genomic DNA of VL3096 as template. The resulting fragments were fused by overlap PCR and transformed into *S. pneumoniae* D39V. Transformants were selected on Colombia agar plates containing, obtaining VL3099.

## Construction of *bgaA::spcR, P3_spv_1159-sfGFP* (VL3100, VL3348)

To insert the *spcR, P3_spv_1159-sfGFP* construct right at the *bgaA* locus, *spcR, P3_spv_1159-sfGFP* was amplified by PCR with OVL2855/OVL2856 using genomic DNA of VL1785 (*CEP::spcR, P3_spv_1159-sfGFP*) as template. Upper and downer fragments were amplified by PCR with OVL1312/OVL2863 and OVL2864/OVL2871 using genomic DNA of D39V, respectively. The three resulting fragments were fused by Golden Gate assembly with BsmBI and transformed into *S. pneumoniae* D39V. Transformed clone was selected by spectinomycin, obtaining VL3100.

To obtain strain VL3348, the *bgaA::spcR, P3_spv_1159-sfGFP* fragment was amplified by PCR with OVL1312/2871 using genomic DNA of VL3100, and transformed into VL1780 to obtain VL3348 (*hlpA::hlpA_hlpA–mScarlet-I, cmR; bgaA::spcR, P3_spv_1159-sfGFP*).

## Construction of *bgaA::spcR, P3_spv_1159-stop-sfGFP* (VL3101, VL3349)

To disrupt translation between *spv_1159* and *sfGFP*, on *bgaA::spcR, P3_spv_1159-sfGFP* construct, single nucleotide mutation was introduced in domain breaking linker between *spv_1159* and *sfGFP*. Upper or downer fragments were amplified by PCR with OVL1312/OVL724 or OVL723/OVL2871 using genomic DNA of VL3096 as template. The resulting fragments were fused each other by overlap PCR and transformed into *S. pneumoniae* D39V. Transformed clone was selected by chloramphenicol, obtaining VL3101.

To obtain dual reporter strain, the *bgaA::spcR, P3_spv_1159-stop-sfGFP* fragment was amplified by PCR with OVL1312/2871 using genomic DNA of VL1786, and transformed into VL1784 to obtain VL3349 (*hlpA::hlpA_hlpA-stop-mScarlet-I, cmR; bgaA::spcR, P3_spv_1159-stop-sfGFP*).

## Construction of *CEP::spcR, P3_spv_1159-sfGFP(inverted)* (VL3346)

To re-introduce *spcR, P3_spv_1159-sfGFP* in inverted direction at *CEP* locus, *spcR, P3_spv_1159-sfGFP* was amplified by PCR with OVL3358/OVL3359 using genomic DNA of VL1785 (*CEP::spcR, P3_spv_1159-sfGFP*) as template. Upper and downer fragments were amplified by PCR with OVL37/OVL3390 and OVL3391/OVL40 using genomic DNA of D39V, respectively. The three resulting fragments were fused by Golden Gate assembly with BsmBI and transformed into *S. pneumoniae* VL1784. Transformed clone was selected by spectinomycin, obtaining VL3346 (*hlpA::hlpA_hlpA-mScarlet-I, cmR; CEP::spcR, P3_spv_1159-sfGFP(inverted)*).

## Construction of *CEP::spcR, P3_spv_1159-stop-sfGFP(inverted)* (VL3347)

To disrupt translation between *spv_1159* and *sfGFP*, on *CEP::spcR, P3_spv_1159-sfGFP(inverted)* construct, single nucleotide mutation was introduced in domain breaking linker between *spv_1159* and *sfGFP*. Upper or downer fragments were amplified by PCR with OVL37/OVL723 or OVL724/OVL40 using genomic DNA of VL3346 as template. The resulting fragments were fused each other

by overlap PCR and transformed into *S. pneumoniae* VL1784. Transformed clone was selected by chloramphenicol, obtaining VL3347 (*hlpA::hlpA_hlpA-stop-mScarlet-I*, *cmR*; *CEP:: spcR*, *P3_spv_1159-stop-sfGFP(inverted)*).

### Construction of *bgaA::spcR, P3_spv_1159-sfGFP(inverted)* (VL3350)
To re-introduce *spcR, P3_spv_1159-sfGFP* in inverted direction at *CEP* locus, *spcR, P3_spv_1159-sfGFP* was amplified by PCR with OVL3358/OVL3359 using genomic DNA of VL1785 (*CEP::spcR, P3_spv_1159-sfGFP*) as template. Upper and downer fragments were amplified by PCR with OVL1312/OVL2863 and OVL2864/OVL2871 using genomic DNA of D39V, respectively. The three resulting fragments were fused by Golden Gate assembly with BsmBI and transformed into *S. pneumoniae* VL1784. Transformed clone was selected by spectinomycin, obtaining VL3346 (*hlpA:: hlpA_hlpA-mScarlet-I ,cmR; CEP::spcR, P3_spv_1159-sfGFP(inverted)*).

### Construction of *bgaA::spcR, P3_spv_1159-stop-sfGFP(inverted)* (VL3351)
To disrupt translation between *spv_1159* and *sfGFP*, on *bgaA::spcR, P3_spv_1159-sfGFP(inverted)* construct, single nucleotide mutation was introduced in domain breaking linker between *spv_1159* and *sfGFP*. Upper or downer fragments were amplified by PCR with OVL1312/OVL723 or OVL724/ OVL2871 using genomic DNA of VL3346 as template. The resulting fragments were fused each other by overlap PCR and transformed into *S. pneumoniae* VL1784. Transformed clone was selected by chloramphenicol, obtaining VL3347 (*hlpA::hlpA_hlpA-stop-mScarlet-I*, *cmR*; *CEP::spcR, P3_spv_1159-stop-sfGFP(inverted)*).

### Construction of *ftsZ-mTurquoise2* (VL3126)
To construct a triple labeled strain, upper or downer fragments were amplified with primer pair OVL452/OVL1921 or OVL1922/OVL1441 using genomic DNA of *ftsZ::ftsZ-mTurquoise2, spcR* strain (Gallay C and Veening JW, unpublished) as template. These fragments were fused by overlap PCR to remove the spcR gene. The fused fragment was used for transformation in D39V and spectinomycin-susceptible clone was selected.

To construct triple labeled strain, *hlpA::hlpA_hlpA-mScarlet-I, cmR* was amplified with OVL43/ OVL46 using genomic DNA of VL1780 and *cep::spcR, P3_spv_1159-msfYFP* was amplified with OVL37/OVL40 using genomic DNA of *cep::spcR, P3_spv_1159-msfYFP (codon-optimized)* strain (Rueff AS and Veening JW, unpublished). These two DNA fragments were transformed into the strain (*ftsZ::ftsZ-mTurquoise2*) and transformed clone was selected by chloramphenicol and spectinomycin, obtaining triple labeled strain VL3126 (*ftsZ::ftsZ-mTurquoise2; hlpA::hlpA_hlpA-mScarlet-I, cmR; cep::spcR, P3_spv_1159-msfYFP*).

### Construction of *ftsZ-stop-mTurquoise2* (VL3127)
To construct triple reporter strain, upper or downer fragments were amplified with primer pair of OVL452/OVL724 or OVL723/OVL1441 using genomic DNA of VL3126 as template. These fragments were fused each other by overlap PCR to introduce stop codon between *ftsZ* and *mTurquoise2*. The fused fragment *ftsZ::ftsZ-stop-mTurquoise2* was transformed in *ftsZ::ftsZ-mTurquoise2* strain and clone that lost mTurquoise2 fluorescence was screened by fluorescence microscopy.

To disrupt translation between *spv_1159* and *msfYFP*, on *CEP::spcR, P3_spv_1159-msfYFP* construct, single nucleotide mutation was introduced in domain breaking linker between *spv_1159* and *msfYFP*. Upper or downer fragments were amplified by PCR with OVL37/OVL724 or OVL723/OVL40 using genomic DNA of VL3126 as template. While, *hlpA-::hlpA-stop-mScarlet-I, cmR* was amplified by PCR with OVL43/OVL46 using VL1784. The resulting two fragments were transformed into *ftsZ:: ftsZ-stop-mTurquoise2* and transformed clone was selected by chloramphenicol and spectinomycin, obtaining VL3128.

### Construction of *recA::recA-mCherry, eryR* (VL361)
*mCherry-eryR* was amplified with RR93/RR94 using VL371 (RR27) (*van Raaphorst et al., 2017*). Upper or downer fragments were amplified by PCR with RR91/RR92 or RR93/RR94 using genomic DNA of D39V as template. The three fragments were assembled using Gibson one-step ISO assembly (*Gibson, 2011*) and transformed into D39V. Transformed clone was selected by erythromycin.

## Construction of R4692 ($\Delta comCDE::trmpR$; $smR^R$; $rifR$; $novR$)

R304 (*smR*; *rifR*; *novR*) (*Chastanet et al., 2001*) strain was transformed with genomic DNA from R4574 (*ΔcomCDE::trmpR*) (*Johnston et al., 2020*) and was selected by trimethoprim, obtaining R4692.

## Luminescence assays of competence development

To monitor competence development, strains containing a transcriptional fusion of the firefly *luc* gene with the late competence gene *ssbB* were used. Cells were pre-cultured in C+Y (pH 6.8) at 37° C to an OD595 nm of 0.2. Right before inoculation, cells were collected by centrifugation (6000 x g for 3 min) and resuspended in fresh C+Y at pH 7.9, which is permissive for natural competence. Luciferase assays were performed in 96-wells plates with a Tecan Infinite 200 PRO illuminometer (TECAN) at 37°C as described before (*Slager et al., 2014*). Luciferin was added at a concentration of 0.45 mg/mL to monitor competence by means of luciferase activity. Optical density (OD595nm) and luminescence (relative luminescence units [RLU]) were measured every 10 min.

## Phase contrast and fluorescence microscopy

Microscopy acquisition was performed using a Leica DMi8 microscope with a sCMOS DFC9000 (Leica) camera and a SOLA light engine (Lumencor) and a 100x/1.40 oil-immersion objective. Images were primarily processed using LAS X (Leica). For snap shot imaging, cells were concentrated 10x by centrifugation (6000x g, 3 min) and 0.5 µL of cells were spotted on 1% agarose/PBS. For time-lapse microscopy, a semi-solid growth surface was prepared with C+Y (pH 7.9) containing 1% agarose in Gene Frame (Thermo Fischer) (*de Jong et al., 2011*). As C+Y medium has some background fluorescence, the C+Y agar pad was pre-exposed on a UV illuminator for 1 min to bleach the background fluorescence.

Phase contrast images were acquired using transmission light with 100 ms exposure for snap shot and 50 ms exposure for time-lapse. Fluorescence was usually acquired with 700 ms exposure for snap shot, and 200–500 ms exposure (17–30% of power from SOLA light engine) for time-lapse using filter settings described below. Time-lapses images were recorded by taking images every 5 or 10 min.

Leica DMi8 filters set used are as followed: mTurquoise2 (Ex: 430/24 nm Chroma, BS: LP 455 Leica 11536022, Em: 470/24 nm Chroma ET470/24 nm or Ex: 430/29 nm Chroma, BS: 455 (450–490) Chroma 69008, Em: 470/26), sfGFP (Ex: 470/40 nm Chroma ET470/40x, BS: LP 498 Leica 11536022, Em: 520/40 nm Chroma ET520/40 m), msfYFP (Ex: 500/20 nm Chroma ET500/20x, BS: LP 520 Leica 11536022, Em: 535/30 nm Chroma ET535/30 m or Ex:495/25 nm Chroma ET495/25x, BS520 (510–560) Chroma 69008, Em: 533/30 nm) and mScarlet-I (Chroma 49017, Ex: 560/40 nm, BS: LP 590 nm, Em: LP 590 nm or Ex: 575/35 nm, BS: 595 (590–670) nm Chroma 69008, Em: 635/70 nm). Microscopy images are available at the BioImages Archive (accession S-BIAD26).

## Quantitative image analysis

For quantitative image analysis of single cells, obtained microscopic images were processed by FIJI software (*Schindelin et al., 2012*). Single-cell segmentation and fluorescence signal intensity measurement were performed by Oufti (*Paintdakhi et al., 2016*). The generated celllist files were analyzed in R (https://www.r-project.org/), using BactMAP (*van Raaphorst et al., 2020*) for statistical analysis and visualization. After celllist file were imported into R, cells were filtered between 0.7–1.2 µm in width length to exclude false events derived from noise or miss-segmentation. Threshold of fluorescence of signal intensity was defined based on negative or positive control for each experiment setting. >500 cells were analyzed at least for each replicate. To exclude the possibility of overlap in detection of fluorescence (particularly mTurquoise2/msfYFP and msfYFP/mScarlet-I) in multi-fragments transformation, we ensured that single transformation experiments did not show any signal in the other channels and this was confirmed by looking at the protein localization patterns.

For generating cell lineage trees from time-lapse imaging, the stacked time-lapse images were processed by FIJI and stabilization between time frames was performed by Huygens (Scientific volume imaging). Single-cell segmentation and fluorescence intensity acquisition were performed by SuperSegger (*Stylianidou et al., 2016*). The resultant data set was analyzed using BactMAP (*van Raaphorst et al., 2020*).

## DNA-binding assays

Analysis of DNA binding was performed in an *endA* mutant background (strain D39V *ssbB::luc (cam) endA::kan*), to favor accumulation of transforming DNA at the surface of competent cells. In wild-type, *endA*+ cells, surface-bound DNA is immediately internalized into the cytosol or degraded otherwise, which makes surface-bound DNA accumulation hard to be visualized as previously shown (*Bergé et al., 2013*). After gently thawing stock cultures, aliquots were inoculated at an OD550 of 0.006 in C+Y medium, supplemented with 20 mM HCl to prevent spontaneous competence development, and grown at 37°C to an OD550 of 0.3. These precultures were inoculated (1/50) in C+Y medium (pH 7.8) and incubated at 37°C. In these conditions, competence developed spontaneously and reached its maximal level in the population after 55–60 min. At 35 min, 1 ml samples were collected and induced, or not, with synthetic CSP (50 ng/ml). At 50 min, these samples were incubated for 5 min with 10 ng of a 285 bp DNA fragment labeled with a Cy3 fluorophore at its 5′ extremities (*Bergé et al., 2013*). Cells were pelleted (3000 x g, 3 min), washed twice in 500 μL C+Y, and resuspended in 20 to 50 μL C+Y medium before microscopy. Two μL of this suspension was spotted on a microscope slide containing a slab of 1.2% C+Y agarose as described previously (*de Jong et al., 2011*).

Phase contrast and fluorescence microscopy were performed with an automated inverted epi-fluorescence microscope Nikon Ti-E/B, a phase contrast objective (CFI Plan Apo Lambda DM 100X, NA1.45), a Semrock filter set for Cy3 (Ex: 531BP40; DM: 562; Em: 593BP40), a LED light source (Spectra X Light Engine, Lumencor), and a sCMOS camera (Neo sCMOS, Andor). Images were captured and processed using the Nis-Elements AR software (Nikon). Cy3 fluorescence images were false colored red and overlaid on phase contrast images. Overlaid images were further analyzed to quantify the number of cells bound with Cy3-labeled DNA. Single cells were first detected using the threshold command from Nis-Elements and cells bound or not to DNA were manually classified using the taxonomy tool.

## Evaluation of transformation frequency using the fluorescence reporter

To quantify the efficiency of transformation with tDNA fragments, reporter cells were pre-cultured in C+Y (pH 6.8) at 37°C to an OD595 nm of 0.2. Right before inoculation, cells were collected by centrifugation (6000 x g for 3 min) and resuspended in fresh C+Y at pH 7.9, adjusted to OD = 0.1. Competence was induced by incubation in the presence of CSP (100 ng/μL) at 37°C for 10 min and then donor tDNA was provided the indicated concentration. After an additional 4 hr of incubation at 37°C for complete cell division to form homo-duplex and maturation of fluorescence proteins, cells were placed on ice to stop cell growth and were directly analyzed by fluorescence microscopy or flow-cytometry.

Donor tDNA was designed in such a way that the single nucleotide mutation is positioned in the middle of the entire fragment so that the left and right homology arms are of equal length. Preparation of the tDNA was performed by PCR using primer pairs indicated in *Supplementary file 2*, using the corresponding parent strain as template. For competition experiments using unrelated tDNA as shown in *Figure 7*, a DNA fragment that has no homology to the pneumococcal genome but is of equal size (5 kb) and GC content (~40% of GC) to the *hlpA-mScarlet-I/spv_1159-sfGFP* fragments, was amplified by PCR from genomic DNA of *E. coli* DH5alpha (*Hanahan et al., 1991*), using primers OVL3527 and OVL3528.

## Flow-cytometry analysis

Cells were collected by centrifugation (6000 x g for 3 min) and resuspended in filtered (0.22 μm) PBS adjusted to a cell density of approximately $1.0 \times 10^5$ to $1.0 \times 10^6$ cells/mL. As encapsulated *S. pneumoniae* D39V cells tend to form chains particularly during competence, we separated cells by bead beating (BioSpec) without any glass beads. At least $>1.0 \times 10^4$ events were analyzed on a Novocyte flow cytometer (ACEA bioscience) harboring 488 nm and 561 nm lasers. Fluorescence filters used were: FITC (Ex: 488 nm, Em: 530/45 nm) for sfGFP and PE.Texas.Red (Ex: 561 nm, Em: 615/20 nm) for mScarlet-I. Obtained raw data were imported and analyzed in R. Non-bacterial particles were excluded by gating the FSC and SSC values. A threshold was determined so that positive events counted in a negative control strain were <1% and validated with both negative (no DNA control) and positive (parent strain without the point mutation) control for each experimental setting.

## Transformation assays between *S. pneumoniae* strains

We used strain R895 as a recipient strain and R4692 and R4574 as donor strains. R895 is a naturally competent derivative of unencapsulated strain R6 and carries a chloramphenicol resistance marker (*Chastanet et al., 2001*). R4692 is unable to develop competence and carries point mutations conferring resistance to streptomycin, novobiocin and rifampicin. R4574 was used as a negative control donor strain. All strains were pre-cultured in C+Y (pH 6.8) at 37°C to an OD550 nm of 0.1. Cells were then collected by centrifugation (6000 x g for 3 min) and concentrated 3-fold in fresh C+Y at pH 7.9. Mixed inoculates containing 30 µL of donor strain R895 and 70 µL of recipient strain (R4574 or R4692) were subsequently spotted onto Petri dishes containing 3% horse blood CAT-agar (1% agar) supplemented with catalase (300 U/mL). Different ratios of donor and recipient were tested (1:1, 1:10 and 3:7) and a 3:7 ratio showed the least HGT variation across experiments. After 4 hr incubation at 37°C, cells were scraped off the plates and separated by sonication in an ultrasonic cleaner (90 s, 50 Hz). Serial dilutions were plated with 4.5 µg/mL of chloramphenicol and with combinations of chloramphenicol plus streptomycin (10 µg/mL) and/or novobiocin (4 µg/mL). Transformation efficiency was calculated by dividing the number of transformants by the total number of viable recipient count. Four independent biological replicates were performed.

## Acknowledgements

We thank Jelle Slager and Arnau Domenech for critically reading this manuscript, two anonymous referees, Juan Carlos Alonso and all members of the Veening lab for stimulating discussions. Work in the Veening lab is supported by the Swiss National Science Foundation (SNSF) (project grant 31003A_172861), a JPIAMR grant (40AR40_185533) from SNSF and ERC consolidator grant 771534-PneumoCaTChER. Work in the Polard lab is supported by the Centre National de la Recherche Scientifique, Université Paul Sabatier, and the Agence Nationale de la Recherche (grants ANR-10-BLAN-1331 and ANR-17-CE13-0031). Jun Kurushima was supported by The Naito Foundation.

## Additional information

### Funding

| Funder | Grant reference number | Author |
| --- | --- | --- |
| Schweizerischer Nationalfonds zur Förderung der Wissenschaftlichen Forschung | 31003A_172861 | Jan-Willem Veening |
| European Commission | 771534- PneumoCaTChER | Jan-Willem Veening |
| Agence Nationale de la Recherche | ANR-10-BLAN-1331 | Patrice Polard |
| Naito Foundation | | Jun Kurushima |
| Schweizerischer Nationalfonds zur Förderung der Wissenschaftlichen Forschung | 40AR40_185533 | Jan-Willem Veening |
| Agence Nationale de la Recherche | EXStasis-17-CE13-0031-01 | Patrice Polard |

The funders had no role in study design, data collection and interpretation, or the decision to submit the work for publication.

### Author contributions

Jun Kurushima, Conceptualization, Data curation, Formal analysis, Investigation, Methodology, Writing - original draft, Project administration; Nathalie Campo, Formal analysis, Investigation, Methodology, Writing - review and editing; Renske van Raaphorst, Resources, Software, Formal analysis, Methodology; Guillaume Cerckel, Investigation; Patrice Polard, Supervision, Funding acquisition, Writing - review and editing; Jan-Willem Veening, Conceptualization, Supervision, Funding acquisition, Writing - original draft, Project administration

## Author ORCIDs

Jun Kurushima (iD) https://orcid.org/0000-0002-8573-6169
Nathalie Campo (iD) https://orcid.org/0000-0003-2917-930X
Renske van Raaphorst (iD) https://orcid.org/0000-0001-7778-5289
Patrice Polard (iD) http://orcid.org/0000-0002-0365-4347
Jan-Willem Veening (iD) https://orcid.org/0000-0002-3162-6634

## Decision letter and Author response

Decision letter https://doi.org/10.7554/eLife.58771.sa1
Author response https://doi.org/10.7554/eLife.58771.sa2

## Additional files

### Supplementary files

- Supplementary file 1. Strains used in this study (MS excel file).
- Supplementary file 2. Oligonucleotides used in this study (MS excel file).
- Transparent reporting form

### Data availability

This work is mainly based upon microscopy and snap shots and movies of most experiments are included in the manuscript and supporting files. Raw microscopy images will be made available at the BioImages Archive (accession S-BIAD26).

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
