## [Decision Letter]

**Acceptance summary:**

This elegant and insightful paper develops a powerful new set of quantitative assays for tracking recombination in single cells to answer fundamental questions about natural transformation in the pathogenic bacterium *Streptococcus pneumoniae*; and thereby, resets paradigms in this research area. The paper shows that any site or orientation with regard to DNA replication can be transformed in pneumococcal cells, including multiple chromosomal insertions; however, there is an intrinsic limitation to the efficiency of recombination, possibly related to the level of off-marker recombination. This limitation likely has implications to pneumococcal genome evolution, affecting exchange of genes that determine virulence and antibiotic resistance.

**Decision letter after peer review:**

Thank you for submitting your article "Unbiased homeologous recombination during pneumococcal transformation allows for multiple chromosomal integration events" for consideration by *eLife*. Your article has been reviewed by three peer reviewers, and the evaluation has been overseen by Petra Levin as Reviewing Editor and Gisela Storz as the Senior Editor. The following individual involved in review of your submission has agreed to reveal their identity: Jason Rosch (Reviewer #1).

The reviewers have discussed the reviews with one another and the Reviewing Editor has drafted this decision to help you prepare a revised submission.

All three reviewers felt that the manuscript was experimentally sound and praised the authors for their use of single cell analysis to tackle the question of why some cells are transformed and others are not in a population of genetically competent Pneumococci. The thoughtful presentation of complicated and extensive data was appreciated by all. Two reviewers were enthusiastic about the study's conclusions regarding bet hedging and the potential for an intrinsic limit on recombination efficiency, reducing the potential for off-marker recombination which, as reviewer #3 notes, might have implications to pneumococcal evolution. At the same time, reviewer #2 had some reservations about the significance of the data in light of previous studies of Pneumococcus and other naturally competent organisms. Most importantly, this reviewer questions whether the finding that only a portion of bacteria incorporate exogenous DNA is a particularly novel and, regardless, whether the saturating DNA concentrations used in the study are representative of a "natural" environment.

Given the discrepancy between the three reviewers with regard to the papers potential impact, we would like to request that you and your co-authors revise the paper, paying special attention to ensuring that the major conclusions are contextual both in terms of previous work and with regard to physiological relevance. I am including reviewer #2's comments in their entirety at the end of this letter to ensure you are aware of all concerns in this regard.

Reviewer #2:

In this work Kurushima et al. use recently developed fluorescent labelling techniques to study natural transformation in the human pathogen *Streptococcus pneumoniae*. Previously, genetic marker analyses have been used to study the different aspect of this process, but with these new techniques the process can now be studied at the single cell level. The authors used the single cell analysis to identify new transformation bottlenecks and tried to determine why some cells are genetically transformed and others are not. Related experiments have been performed in the past using classic genetics and Kurushima et al. were able to confirm these studies. In that sense, in my opinion, the novelty is limited and no important new molecular insights are provided. They found that the number of cells that are ultimately transformed is plateauing at approximately 50%, despite the fact that most cells bind DNA. This is partially the result of the heteroduplex formed after recombination followed by separation by strand replication, combined with the fact that the DNA binding sites on cells are limited so that there is a competition between DNA markers at saturating DNA concentrations. The authors argue that this mechanism entails a "fail-safe strategy for the population as half of the population generally keeps an intact copy of the original genome". I find this conclusion far-fetched for two reasons. Firstly, the DNA recombination event followed by DNA replication will automatically assure that only half the population will inherit the mutation, and to speak of a strategy implies that the organism has specifically evolved this system, but we are dealing here with a well-known and general recombination system found in many organisms that will generally result in a 50/50 distribution. Maybe more importantly, under natural conditions it is highly unlikely that cells encounter saturating levels of tDNA. In their experiments the authors use 3.2 nM DNA for transformation. If my calculation is correct, this would amount to 19xE11 DNA molecules per ml, which seems a bit high when assuming tDNA comes from lysed bacteria. In nature, this number will be much (much) smaller therefore there is no need for the bacterium to come up with a dedicated strategy to assure that not all cells in a population are being transformed. Finally, the results are very well presented and the paper makes easy reading.

---

## [Author Response]

Reviewer #2:In this work Kurushima et al. use recently developed fluorescent labelling techniques to study natural transformation in the human pathogen Streptococcus pneumoniae. Previously, genetic marker analyses have been used to study the different aspect of this process, but with these new techniques the process can now be studied at the single cell level. The authors used the single cell analysis to identify new transformation bottlenecks and tried to determine why some cells are genetically transformed and others are not. Related experiments have been performed in the past using classic genetics and Kurushima et al. were able to confirm these studies. In that sense, in my opinion, the novelty is limited and no important new molecular insights are provided. They found that the number of cells that are ultimately transformed is plateauing at approximately 50%, despite the fact that most cells bind DNA. This is partially the result of the heteroduplex formed after recombination followed by separation by strand replication, combined with the fact that the DNA binding sites on cells are limited so that there is a competition between DNA markers at saturating DNA concentrations. The authors argue that this mechanism entails a "fail-safe strategy for the population as half of the population generally keeps an intact copy of the original genome". I find this conclusion far-fetched for two reasons. Firstly, the DNA recombination event followed by DNA replication will automatically assure that only half the population will inherit the mutation, and to speak of a strategy implies that the organism has specifically evolved this system, but we are dealing here with a well-known and general recombination system found in many organisms that will generally result in a 50/50 distribution. Maybe more importantly, under natural conditions it is highly unlikely that cells encounter saturating levels of tDNA. In their experiments the authors use 3.2 nM DNA for transformation. If my calculation is correct, this would amount to 19xE11 DNA molecules per ml, which seems a bit high when assuming tDNA comes from lysed bacteria. In nature, this number will be much (much) smaller therefore there is no need for the bacterium to come up with a dedicated strategy to assure that not all cells in a population are being transformed. Finally, the results are very well presented and the paper makes easy reading.

We thank the reviewer for hinting to us to look at more environmentally relevant scenarios. To this end, we performed a set of new experiments (new Figure 8) in which we grew two pneumococcal strains together on agar plates in which one of the strains cannot become competent and carries SNPs conferring streptomycin and novobiocin resistance, respectively. In this scenario, horizontal gene transfer occurs via lysis of the recipient and by transformation of the then liberated environmental chromosomal DNA (new Figure 8A). One could argue this experiment more closely mimics the natural polymicrobial biofilm-like niche pneumococci encounter in the human nose, compared to our in vitro experiments using PCR products as donor DNA. As shown in new Figure 8B, also in this experimental setup, transformation is very efficient, up to 5.70 x 10^-2^ (~0.05% of recipient cells are transformed) and markers transform independent of each other as demonstrated by our single cell analysis. We think this new experiment underscores that competence-based transformation is a main driver for pneumococcal genome evolution. We like to note that we do not think that the fail-safe has evolved as a dedicated strategy, but is rather a consequence of the recombination mechanism. This was mentioned in the Discussion as being a ‘spandrel effect’ and we have now further clarified this in the revised manuscript and changed fail-safe mechanism to fail-safe scenario.